# Crafting a Personalized Prognostic Model for Malignant Prostate Cancer Patients Using Risk Gene Signatures Discovered through TCGA-PRAD Mining, Machine Learning, and Single-Cell RNA-Sequencing

**DOI:** 10.3390/diagnostics13121997

**Published:** 2023-06-07

**Authors:** Feng Lyu, Xianshu Gao, Mingwei Ma, Mu Xie, Shiyu Shang, Xueying Ren, Mingzhu Liu, Jiayan Chen

**Affiliations:** 1Department of Radiation Oncology, Peking University First Hospital, Beijing 100034, China; lyufeng2018@bjmu.edu.cn (F.L.); 15810160120@163.com (M.M.); xiemu0312@126.com (M.X.); ssy970726@163.com (S.S.); xy_ren1031@126.com (X.R.); liumingzhu125@126.com (M.L.); sunshinecan2011@sina.com (J.C.); 2First Clinical Medical School, Hebei North University, Zhangjiakou 075000, China

**Keywords:** prostate cancer, prognostic model, immunotherapy, machine learning, single-cell RNA-sequencing

## Abstract

Background: Prostate cancer is a significant clinical issue, particularly for high Gleason score (GS) malignancy patients. Our study aimed to engineer and validate a risk model based on the profiles of high-GS PCa patients for early identification and the prediction of prognosis. Methods: We conducted differential gene expression analysis on patient samples from The Cancer Genome Atlas (TCGA) and enriched our understanding of gene functions. Using the least absolute selection and shrinkage operator (LASSO) regression, we established a risk model and validated it using an independent dataset from the International Cancer Genome Consortium (ICGC). Clinical variables were incorporated into a nomogram to predict overall survival (OS), and machine learning was used to explore the risk factor characteristics’ impact on PCa prognosis. Our prognostic model was confirmed using various databases, including single-cell RNA-sequencing datasets (scRNA-seq), the Cancer Cell Line Encyclopedia (CCLE), PCa cell lines, and tumor tissues. Results: We identified 83 differentially expressed genes (DEGs). Furthermore, WASIR1, KRTAP5-1, TLX1, KIF4A, and IQGAP3 were determined to be significant risk factors for OS and progression-free survival (PFS). Based on these five risk factors, we developed a risk model and nomogram for predicting OS and PFS, with a C-index of 0.823 (95% CI, 0.766–0.881) and a 10-year area under the curve (AUC) value of 0.788 (95% CI, 0.633–0.943). Additionally, the 3-year AUC was 0.759 when validating using ICGC. KRTAP5-1 and WASIR1 were found to be the most influential prognosis factors when using the optimized machine learning model. Finally, the established model was interrelated with immune cell infiltration, and the signals were found to be differentially expressed in PCa cells when using scRNA-seq datasets and tissues. Conclusions: We engineered an original and novel prognostic model based on five gene signatures through TCGA and machine learning, providing new insights into the risk of scarification and survival prediction for PCa patients in clinical practice.

## 1. Introduction

Prostate cancer (PCa) is the second most predominant and fifth most lethal tumor affecting men worldwide and is a serious threat to men’s health. In 2020, roughly 1.4 million newly diagnosed patients with PCa were recorded [1]. PCa is principally considered a comparatively indolent and slowly progressing neoplasm that clinically represents a strongly localized stage for the radical treatment of menthids. Although patients with PCa have a markedly long overall survival (OS), the 5-year OS of PCa patients exceeds 90% for localized tumors [2]. Radical prostatectomy, androgen deprivation therapy (ADT), and radiation therapy (RT) can achieve favorable treatment outcomes. Unfortunately, patients with some highly malignant prostate neoplasms, such as those with a high Gleason score (GS) (GS ≥ 9), have been exposed as insensitive to ADT in current clinical trials. The OS was found to be similar between the group receiving ADT and the group that did not receive ADT, with the OS approaching 80% [3,4]. In addition, other patients with CRPC were resistant to multiple clinical therapies, such as radiotherapy, ADT, and chemotherapy [5,6,7]. The prognosis of PCa patients with high GS was poor: a bottleneck for clinical diagnosis and treatment. The characteristics of expression profiles and biological behavior of GS 9-–10 patients with PCa remain unclear. Therefore, a novel risk model based on specific molecular profiles and clinical data is needed for early clinical detection and the identification of PCa patients, especially those with GS 9–10. 

Current clinical risk stratification tools and models for PCa patients include the TNM stage, the prostate-specific antigen (PSA) level, and GS [6]. Owing to the heterogeneity of PCa tumors and the personalization of patients [8], conventional risk stratification methods and models cannot stratify the detailed risk groups of patients and predict survival. Owing to the rapid development and advancement of molecular biology [9,10], the emergence of novel molecular biomarkers has provided a new approach for risk stratification and model construction for PCa patients, such as the androgen receptor (AR), prostate-specific membrane antigen (PSMA), and other molecular chaperones [11]. Additionally, there are many commercially available scoring panels, such as the Decipher score, Oncotype DX, Prolaris, and Stockholm 3 (STHLM3) model [12]. Decipher scores, including 22 RNA biomarkers and clinical variables, could prognosticate a 10-yr distant metastasis rate after radical prostatectomy (RP), whose area under the curve (AUC) was 0.81 [13,14]. The STHLM3 model is a clinical tool for diagnosis according to the individual risk that mixes plasma biomarkers, such as PSA, free PSA, intact PSA, 232 genetic polymorphism sites, and some clinical variables (e.g., age, disease and family history, a prior biopsy of the prostate, and prostate examination), with a higher accuracy (AUC = 0.74) than other tools in the identification of GS ≥ 7 high-risk PCa [15]. These risk stratification tools were generated based on several PCa tissue samples and involved numerous genes that were more laborious to clinically detect. In addition, these risk stratification tools and models do not predict immune cell infiltration or treatment response to immune checkpoint inhibitors (ICIs) for PCa. 

In our investigation, we operated the differentially expressed gene (DEGs) analysis of PCa patients and integrated functional enrichment analysis, engineering a prognostic model and OS nomogram based on risk gene signatures to aid clinical decision-making, and assessed the correlation between immune infiltration and risk stratification. We aimed to identify the most impactful features among the risk factors for PCa patients’ prognosis by utilizing machine learning techniques in a rigorously written medical journal style for accuracy of expression. Moreover, we validated the model via the International Cancer Genome Consortium (ICGC) dataset. We further used real-time quantitative polymerase chain reaction (RT-qPCR), the single-cell RNA-sequencing (scRNA-seq) datasets, and The Human Protein Atlas (HPA) database to assess the cellular distribution and expression levels for these risk gene signatures. Our study revealed new patterns and perspectives for the risk stratification of PCa patients in current clinical management. Of note, our study has provided a novel perspective for the early clinical recognition of high GS patients and offers novel diagnostic tools for timely intervention in these patients.

## 2. Materials and Methods

### 2.1. Sequencing Data Access

RNA-seq data in this study were procured from The Cancer Genome Atlas Program (TCGA) Prostate Adenocarcinoma Database (TCGA-PRAD, https://www.cancer.gov/about-nci/organization/ccg/research/structural-genomics/tcga, accessed on 1 November 2022). Overall, 499 PCa patients were enrolled, including 357 patients with GS 6–8, 142 patients with GS 9–10, and 52 normal controls (from The Genotype-Tissue Expression) (Appendix A). The survival (*n* = 405) and death (*n* = 94) groups were ranked according to the time of disease progression (biochemical recurrence). 

### 2.2. Difference Expression of Gene Analysis

The total RNA-seq datasets of TCGA-PRAD were divided into three individual comparable cohorts: PCa patients (*n* = 499) vs. normal tissue (*n* = 52), PCa patients with GS 9–10 (*n* = 142) vs. PCa patients with GS 6–8 (*n* = 357), and PCa patients with rapid disease progression (*n* = 94) vs. PCa patients with slow disease progression (*n* = 405). For each of the above individual comparable cohorts, the “DeSeq2” package (version 1.26.0) in R software (version 3.6.3; the following software versions used for R were all the same) was used for the analysis of DEGs. Adjusted *p* values < 0.05 and log2|Fold Change| > 1 were used to identify the DEGs.

### 2.3. Gene Enrichment Analysis

Gene ontology (GO) enrichment analysis and Kyoto Encyclopedia of Genes and Genomes (KEGG) pathway analysis were implemented to assess the malignant biologic behavior of patients with GS 9–10 and the genes of intersection among the three cohorts using the package “cluster profiles” in R software. A cut-off value of *p* < 0.05 was established to select significantly enriched signaling transduction pathways. Gene set enrichment analysis (GSEA) was performed to explore the possible pathways and biological processes. A false discovery rate (FDR) < 0.25 and p.adjust < 0.05 were set as the threshold for meaningful enrichment. The “clusterProfiler” package in R software (version 3.6.3) was used to complete the enrichment analysis.

### 2.4. Construction of the Protein–Protein Interaction (PPI) Network 

We investigated the PPI networks of DEGs in PCa patients with GS 9–10 using the Metascape online database (http://metascape.org, accessed on 1 November 2022). Hub genes were selected and displayed based on the molecular complex detection (MCODE) algorithm.

### 2.5. Construction and Validation of the Prognostic Model Based on the Risk Gene Signatures

LASSO regression was carried out to determine the regression coefficients of the 83 selected DEGs in the intersection parts. Thereafter, five genes were chosen for the arrangement of the subsequent risk formulation. The risk score was calculated using the formula that follows: Risk Score=expression of Gene 1×b 1+expression of Gene 2×b 2+…+expression of Gene n×b n

Thereafter, we used TCGA-PRAD and separated the patients into low- and high-risk groups following the median risk score. Kaplan-Meier (KM) curves were generated to calculate the progression-free survival (PFS) and distinguish between the low-risk and high-risk groups; log-rank P was used for statistical testing. To estimate the sensitivity and specificity of this model, the “SurvivalROC” package in R software was employed to plot the receiver operating characteristic (ROC) curves for analysis. *p* < 0.05 was considered to represent statistical significance. Finally, we validated our risk model using the PFS and expression profiles of five risk genes from 25 PCa patients in the International Cancer Genome Consortium (ICGC) database. The same data analysis and evaluation methods were employed as previously described.

### 2.6. Construction of the Univariable COX Regression Model and Nomogram

The age, N stage, PSA (ng/mL), GS, and expression levels of IQGAP3, KIF4A, TLX1, KRTAP5-1, and WASIR1 of PCa patients in TCGA-PRAD were employed as the variables for univariable COX regression and were used to analyze the hazard ratio (HR). The above variables were selected for inclusion in the nomogram, and the “rms” and “survivor” packages in R software were used to anticipate the OS of PCa patients. The C-index was also utilized to assess the consistency of the nomogram model.

### 2.7. Machine Learning Model

We partitioned TCGA-PRAD patients into training, testing, and validation sets in a 6:2:2 ratio. We then utilized machine learning models, including nonlinear regression, random forest, and XGBoost, to examine the impact of risk factors on the progression-free interval (PFI) of PCa patients. To optimize model hyperparameters such as the maximal number of samples in the leaf and tree depth, we employed RandomizedSearchCV and GridSearchCV from the Python sci-kit-learn library with 5-fold cross-validation and Bayesian optimization. Additionally, we examined the feature importance and interactions using SHAP (Python, version 3.9.2).

### 2.8. Correlation Analysis of Immune Infiltration and Genes Associated with ICIs

We used the proportion of the immune and cancer cell (EPIC) tool, an mRNA-based prediction algorithm, to estimate the relevance between different risk groups and immune cell infiltration. Some immune-checkpoint-relevant points, such as CTLA4, CD274, SIGLEC15, TIGIT, PDCD1, HAVCR2, PDCD1LG2, and LAG3, were selected for this evaluation. Additionally, the values for these genes’ expressions were extracted for additional study. The “ggplot2” and “pheatmap” packages in the R software were used for visualization.

### 2.9. Verification of Risk Gene Signature Expression Levels

We obtained gene expression matrices of normal prostate cancer epithelia and PCa cell lines from the Cancer Cell Line Encyclopedia (CCLE) dataset (https://portals.broadinstitute.org/ccle/about, accessed on 20 May 2023). The results were visualized using “ggplot2” packages. The RWPE-1 cell line, derived from normal human prostate epithelial cells, and six human PCa cell lines (VCaP, LNCaP, C4-2, 22Rv1, DU145, and PC-3) were sourced either from the American Type Culture Collection (ATCC, USA) or Procell (Wuhan, China). In the following experiments, the identification of all cell lines was conducted using short tandem repeats. RWPE-1 cells were maintained in a K-SFM medium (Invitrogen, Carlsbad, CA, USA), while the other cancer cell lines were cultured in RMPI-1640 (C11875500BT, Gibco, Carlsbad, CA, USA) or DMEM (C11995500BT, Gibco, Carlsbad, CA, USA) with 10% FBS (04-001-1ACS, Bioind Ltd., Nes Ziona, Israel) and 1% penicillin and streptomycin (15140-122, Gibco, Carlsbad, CA, USA) at 37 °C with 5% CO_2_. The total RNA was extracted using the TRIzol reagent (15596026, Invitrogen, Carlsbad, CA, USA) as directed by the manufacturer. A high cDNA capacity kit was applied to reverse transcription (4368814, Applied Biosystems Inc., Foster City, San Francisco Bay Area, USA). The mRNA expression of the associated genes was evaluated using ABI 7500 with an SYBR Green Master Mix (A25741, Applied Biosystems Inc., Foster City, San Francisco Bay Area, USA). The primer sequences are listed in Appendix A. The HPA database (https://www.proteinatlas.org, accessed on 1 November 2022) was consulted to corroborate the protein expression levels of the identified risk genes.

### 2.10. Single-Cell RNA Sequencing Data Analysis

For this investigation, we employed single-cell RNA sequencing (scRNA-seq) data obtained from the Gene Expression Omnibus (GEO), with accession number 141445 [16]. We employed the “Seurat” package to filter out low-quality cells and implement standard data preprocessing procedures for gene counts, cell numbers, and mitochondrial content percentage calculations. Our filtering criteria required the detection of at least three cells per gene and no fewer than 200 detected genes per cell. After conducting sample quality control, batch correction, and the identification of highly variable genes, we applied the UMAP dimensionality reduction technique for cell clustering and annotated this based on their markers. Finally, we analyzed differential gene expression across various cell types, including immune cells.

### 2.11. Statistical Analysis

All analysis methods and R packages were R Foundation for Statistical Computing creations. For data that did not fit a normal distribution, nonparametric tests were utilized. In the examination of the correlation between the prognostic model and immunotherapy-associated signs, the significance of two groups was determined using the Wilcox test, and the significance of three or more groups was determined using the Kruskal–Wallis test.

## 3. Results

### 3.1. Analysis and Screening of Differentially Expressed Genes (DEGs) of PCa Patients in Three Cohorts

Based on TCGA-PRAD, differential gene profiles were separated into three cohorts: PCa vs. normal tissues, PCa patients with GS 9–10 vs. GS 6–8, and PCa patients with rapid disease progression vs. those with slow disease progression (Figure 1A,B). Through a comparison between tumor and normal tissues, we identified 6669 DEGs, consisting of 3692 upregulated and 2977 downregulated genes, as depicted in Figure 1C. Compared with patients exhibiting GS 6–8, those with GS 9–10 exhibited 865 DEGs, whereby 457 were upregulated and 408 downregulated (Figure 1D). Compared to individuals displaying slow disease progression, patients with PCa who experienced rapid disease progression demonstrated alterations in 323 DEGs, with 64 of them showing upregulation and 259 downregulation, as shown in Figure 1E. The overlap among the genetic information from these three parts revealed a total of 83 genes that were commonly affected, displayed in Figure 1F.

### 3.2. Selection of the Prognostic Genes and Analysis of the Clinical Correlation

We found that 83 genes in the three cohorts may have affected the patients’ prognosis. At that point in time, the decipher scores comprised 22 RNA biomarkers to predict the prognosis of PCa. We screened the DEGs and selected fewer genes to construct the prognostic model to achieve more efficiency and accuracy. To assess the correlation between the expression of 83 genes and the prognosis of patients with PCa, LASSO regression was initially conducted. As shown in Figure 2A,B, WASIR1, KRTAP5-1, TLX1, KIF4A, and IQGAP3 were selected for further investigation, as they were found to markedly influence the survival of PCa patients. We proceeded to investigate the correlation between the expression profiles of the aforementioned genes and clinical characteristics in PCa patients. Notably, elevated expression levels of WASIR1, KRTAP5-1, TLX1, KIF4A, and IQGAP3 were linked with more advanced T stage, N stage, higher PSA values, and GS, as demonstrated in Figure 2C–F. These observations suggest that these genes could significantly contribute to PCa development and progression.

### 3.3. Enrichment Analysis of the 83 Genes from Three Cohorts and Prediction of Malignant Biological Behavior in PCa Patients with GS 9–10

We conducted the enrichment analyses of GO/KEGG pathways using 83 genes, which demonstrated their involvement in arachidonic acid metabolism and hormonal regulation, among others (Figure 3A,D). Patients with a GS of 9-10 had a more advanced malignancy grade, and recent clinical trials demonstrated the need for comprehensive therapeutic care in this patient group. Hence, it is imperative to comprehend the malignant biological behavior involved in the disease progression of individuals with GS 9–10, with the purpose of discovering novel therapeutic targets. Functional enrichment analyses were carried out using GO and KEGG. The biological procedures of the cell cycle, bile secretion, steroid hormone biosynthesis, receptor-ligand activity, and hormone activity, which were revealed by enrichment analysis, were identified as the main driving factors for the malignant transition of PCa patients with GS 9–10. Further analysis using GSEA revealed that the biological processes of the cell cycle, cell mitosis, and the Rho signaling pathway were highly related to GS 9–10 patients (Figure 3B,C). 

### 3.4. Construction of the PPI Network for PCa Patients with GS 9–10

To gain more insights into the biological function and behaviors of DEGs in PCa patients with GS 9–10, we constructed PPI network maps for these 865 DEGs. The results of the Metascape enrichment analysis were similar to those of GO/KEGG and GSEA in Figure 3. The results of enrichment analysis revealed that biological behaviors such as cell cycle regulation and androgen synthesis might be key events in GS 9–10 prostate cancer patients. Additionally, as shown in Figure 4C, the hub genes suggest that KIF4A and IQGAP3 may be the key genes regulating GS 9–10 prostate cancer patients.

### 3.5. Construction and Validation of the Prognostic Models Based on Risk Gene Signatures

As shown in Figure 2, we examined risk gene signatures and subsequently utilized a univariate COX regression model to assess the HRs. The T stage, N stage, serum PSA levels, GS, and expression levels of IQGAP3, KIF4A, and KRTAP5-1 were identified as risk factors for the progression-free interval (PFI) (Figure 5A). Therefore, we constructed a nomogram to predict OS based on clinical variables and signatures. The OS nomogram displayed excellent predictive performance based on discrimination and calibration (Figure 5B), with a C-index of 0.823 (95% CI, 0.766–0.881). Moreover, the calibration curve exhibited favorable agreement between the anticipated outcomes of the nomogram and the observed data (Figure 5C).

As PCa has a long disease history and is associated with a markedly long OS time, monitoring disease progression in PCa patients via imaging progression or biochemical failure is very meaningful for clinical surveillance. Accordingly, we built a prognostic model for PCa to predict PFS based on the risk gene signatures. In addition to considering these clinical variables, we constructed a prognostic model utilizing LASSO regression and risk gene signatures. The following formula was used to determine the risk scores:$$Risk\;score=\left(0.6086\right)\times WASIR1+\left(0.3736\right)\times KRTAP5-1+\left(0.6149\right)\;\times TLX1+\left(0.1071\right)\times KIF4A+\left(0.1945\right)\times IQGAP3$$

Based on our research, the implementation of the five-gene prognostic model resulted in a more accurate differentiation between patients with high and low risks for PCa. HR was found to be 4.117 (*p* < 0.0001). Furthermore, the area under the ROC curve for the prediction of PFS in PCa patients at 3, 5, and 10 years was 0.737, 0.647, and 0.788, respectively, as illustrated in Figure 6D. To test the predictive power of our model, we validated it using the ICGC dataset. The risk scores and survival status of PCa patients in the ICGC database, as indicated by the five-gene signature score shown in Figure 6E, were consistent with the expression trend of risk factors and TCGA-PRAD. Moreover, this high-risk group demonstrated a poorer PFS (Figure 6F, *p* < 0.05). The AUC values for predicting PFS at 2, 3, and 4 years were 0.942, 0.759, and 0.576, respectively (Figure 6G), indicating good predictive performance of the model in external datasets.

### 3.6. Using a Machine Learning Model to Analyze the Impact of Risk Molecules’ Features on Disease Progression in PCa Patients

To further explore the influence and interaction between input features and the model, we employed SHAP in a proficient machine learning approach to evaluate the influence of these factors on PFI among PCa patients. Our findings indicated that the features of WASIR1 and KRTAP5-1 played significant roles in the model (as shown in Figure 7A,B) and as demonstrated by the interaction and decision plots in Figure 7C,D. This highlights the importance of WASIR1 and KRTAP5-1 as potential references for predicting survival outcomes in PCa patients, which were further verified in conjunction with the results of LASSO regression.

### 3.7. Correlation Analysis of Risk Prognostic Model and Immune Treatment Response

Immunotherapy has led to dramatic success in bladder, lung, and esophageal cancers; however, PCa has a low tumor mutational burden (TMB) and does not benefit markedly from immunotherapy [17,18,19,20]. Recent studies revealed increased TMB and microsatellite instability (MSI) in CRPC, and some recent clinical trials relating to the immunotherapy exploration of PCa are ongoing [21,22]. Therefore, to enhance the effectiveness of immunotherapy for PCa, specific populations must be filtered to benefit from immunotherapy. We proceeded to determine whether the response to treatment and immune cell infiltration could be associated with this five-gene predictive model. The investigation revealed that the risk score calculated using this five-gene prognostic model had a negative correlation with CD4+ and CD8+ T-cell infiltration and a positive correlation with macrophage infiltration (Figure 8A). Further, the expression levels of IQGAP3 were associated with immune checkpoint-related gene expression (Figure 8B), such as HAVCR2 and CTLA4, which were the targets of ICIs used in the clinic. The expression of KRTAP5-1 and KIF4A was also correlated with the expression of HAVCR2, CTLA4, and LAG3 (Figure 8C), suggesting that this prognostic model may provide a basis for clinical PCa decision-making.

### 3.8. Analysis of Risk Factor Expression in Different Cellular Subpopulations of PCa Tissue

To clarify the expression profiles of various cell types within the PCa tissue, we performed an analysis on the GSE 141445 dataset. As depicted in Figure 9A, our cell grouping results were presented along with markers identified by the clustering method shown in Figure 9B. It is worth noting that while the expression levels of the five risk factors mentioned in this dataset were relatively low, they still exhibited different distribution patterns. Specifically, KIF4A was found to be expressed in tumor cells, T cells, monocytes, and fibroblasts. IQGAP3 was expressed in tumor cells, T cells, and monocytes, while WASIR1 was primarily expressed in mast cells (Figure 9C and Appendix A). Moving forward, as single-cell RNA sequencing methods continue to improve, we plan to integrate additional datasets to further refine our understanding of the expression distributions of these important risk factors.

### 3.9. Assessment of Risk-Associated Gene Signature Expression Patterns in PCa Specimens and Cultures

We subsequently validated the expression differences of risk genes in prostatic epithelial cells (PrEC LH) and a panel of prostate cancer cell lines (VCaP, MDA PCa 2b, DU145, LNCaP, 22Rv1, PC-3, NCI-H660) using the CCLE database (Figure 10A). Interestingly, these genes were found to be upregulated in NCI-H660, a neuroendocrine PCa (NEPC) cell line, suggesting that they may serve as biomarkers for the transformation to NEPC.

After conducting an integrated bioinformatics analysis, we verified the expression of high-risk genes in both PCa cells and patients. We used another prostatic epithelial cell line, RWPE-1. Interestingly, compared to this normal cell line, the expression levels of WASIR1, KRTAP5-1, KIF4A, and IQGAP3 in the PCa cell lines were inconsistent with prior findings (Figure 10B–F). However, we observed the upregulated expression of KRTAP5-1 and TLX1 in all PCa cells (Figure 10C,D). Further analysis at the protein level confirmed that KIF4A and IQGAP3 were upregulated in tumor tissues instead of normal tissues from the HPA database (Figure 10G). These discrepancies could be due to the non-representative nature of cell lines for solid tumor tissues and the influence of hormones and AR. 

## 4. Discussion

In current clinical PCa treatment regimens, the treatment of localized PCa is very impactful, and this prognosis is favorable, with an overall 5-year OS surpassing 90%. However, high-GS patients remain a clinical challenge and a significant source of poor prognosis for affected patients. Conventional treatment options for PCa include surgery, radiotherapy, ADT, and chemotherapy, yet resistance to these therapies is commonly observed in high-GS patients [23]. This highlights the need for improved therapeutic strategies and risk-stratification tools in this patient population to optimize clinical outcomes. Owing to the poor systemic condition of some older patients, these patients are not ideal candidates for traditional treatment methods. The advent of immunotherapy has led to new therapeutic alternatives for PCa. However, the population benefiting from immunotherapy must be further filtered to optimize the outcomes.

Regarding the distinctive profiles of high-GS PCa patients, we identified WASIR1, KRTAP5-1, TLX1, KIF4A, and IQGAP3 as novel biomarkers associated with malignancy in PCa and built an inventive prognostic model to estimate the OS and PFS. Currently, molecular subtyping is the focus and the most effective biomedicine tool to facilitate personalized risk stratification and distinguish patients with the poorest prognosis [9,11]. For example, 17 gene expression classifiers were included in OncotypeDX, and 22 gene expression classifiers were included in the Decipher score [12]. The hallmarks of this tumor were characterized by uncontrolled cell proliferation [24,25], making the establishment of prognostic models based on molecular markers of programmed cell death and physiological process a hot topic in current risk model explorations. For example, predicting models based on crucial regulatory molecules of cuproptosis [26], steroid hormone biosynthesis [27], and m6A methylation [28] are well established, and their correlation with immune cell infiltration can also be analyzed. Disulfidptosis is a newly discovered form of programmed cell death, which is characterized by the inhibition of intracellular disulfide reductants and the significant enhancement of cell death caused by SLC7A11 under glucose starvation conditions, in contrast to the significant increase in cell death induced by thiol oxidation reagents such as diamide [29]. The key regulatory molecules of disulfidptosis may have a predictive value in the prognosis of PCa survival and require further study in future research. 

Other trends suggest that future studies could reduce the number of risk factors and more accurately calculate risk scores to predict prognosis and survival. Ning Shao et al. constructed a risk stratification model to forecast OS and immune cell infiltration in PCa using six genes: ZNF467, SH3RF2, PPFIA2, MYT1, TROAP, and GOLGA7B [30]. In our prognostic model based on five genes, we also used this model to observe and predict immune cell infiltration. Further analysis revealed that the risk genes IQGAP3, KRTAP5-1, and KIF4A affected the expression of genes in relation to ICIs in tumor tissues, which may affect the treatment response to ICIs. Additional functional experiments are needed to test this hypothesis. Alhasan A.H. et al. developed a risk-prognostic model for very high-risk PCa patients by measuring the serum expression levels of miR-200c, miR-605, miR-135a, miR-433, and miR-106a, which are non-coding RNA molecules. The accuracy rate of this model was found to be approximately 89% [31]. In our signature, WASIR1 is also a type of long non-coding RNA molecule. The choice of risk factors in these risk prognostic models, either coding or non-coding RNA molecules, is a hot topic and trend in disease risk stratification owing to the future application of liquid biopsies for non-coding RNA molecules in clinical practice [32]. In addition to molecular types, our study utilized an optimized machine learning model to analyze the survival effect of prognostic molecules on patients. While some studies have employed machine learning to screen for prognostic molecules, our future research may also utilize advanced algorithms in machine and deep learning to identify molecules that are more in line with reality.

In our study, we employed IQGAP3 as an important prognostic factor. IQGAP3 constitutes a family of GTPase-activating proteins featuring IQ motifs and multiple structural domains [33,34]. At present, some studies have revealed that molecules from this family are essentially tumor-associated antigens and are considered candidate biomarkers for malignancy, playing a well-defined functional role in tumorigenesis [35,36]. IQGAP3 was first isolated in 2007 [37]; moreover, research has demonstrated that IQGAP3 is directly associated with cell proliferation, differentiation, and intracellular signal transduction pathways. In gastric cancer, the IGGAP3/Ras axis is necessary for maintaining the homeostasis of epithelial proliferation and the repair processes, and its aberrant expression can be associated with gastric cancer [38]. IQGAP3 is associated with tumor treatment response and tolerance in other types of tumors. Correlations were observed between IQGAP3 expression levels and sensitivity to olaparib in ovarian cancer cells [39]. Additionally, upregulated IQGAP3 expression in individuals with breast cancer has been correlated with a highly unfavorable clinical outcome and increased radioresistance [40]. Nonetheless, relatively few studies have investigated the function of IQGAP3 in PCa. Surprisingly, in contrast to the Decipher score, one of the risk factors in our model, IQGAP3, was implicated in carcinogenesis and the development of PCa. Although the statistical models employed in this analysis incorporated IQGAP3 expression levels as a variable in the risk assessment, machine learning algorithms, and prognosis of PCa patients, the precise mechanistic role of IQGAP3 in tumor initiation, progression, metastasis, and response to treatment remains elusive. Thus, an urgent need exists for further investigations to elucidate the functional significance of IQGAP3 in PCa.

There has been a popular focus on exploring novel therapies for PCa, particularly immunotherapy, in recent years. The FDA-approved initial drug for PCa immunotherapy, Sipuleucel-T, was indicated for asymptomatic or minimally symptomatic metastatic castration-resistant prostate cancer (mCRPC) [41]. ICIs have achieved tremendous success as treatments for many malignant tumors, such as lung and esophageal cancers [19,42,43]. The efficacy of anti-programmed death 1 (PD-1) regimens for treating advanced PCa was evaluated in KEYNOTE-199 and KEYNOTE-288 trials [22,44]. According to the outcomes observed, individuals diagnosed with metastatic castration-resistant prostate cancer (mCRPC) and who were unresponsive to standard therapies exhibited an objective response rate (ORR) of merely 5% in the KEYNOTE-199 trial, even though they tested positive for programmed death ligand 1 (PD-L1) and had a suboptimal survival duration spanning from 7.9 to 14.1 months. Thus, the appropriate population of patients that benefited from immunotherapy for PCa had to be further screened. To address this dilemma, PD-1/PD-L1 expression levels using IHC staining served as a commonly used evaluation metric; however, its predictive value for PD-1/PD-L1 is tumor type-dependent, and PD-1/PD-L1 negative patients also responded to immunotherapy [45]. Accordingly, current immunotherapy modalities for PCa have various limitations. We constructed a risk prognostic model based on five functional molecules, namely WASIR1, KRTAP5-1, TLX1, KIF4A, and IQGAP3, to evaluate immune cell infiltration in PCa tissues based on risk scores. Further research revealed that IQGAP3, KIF4A, and KRTAP5-1 influenced the expression of genes associated with immunological checkpoints, such as CTLA4, HAVCR2, and LAG3. After analyzing the single-cell dataset, it was discovered that IQGAP3, KIF4A, and WASIR1 were expressed in immune cells, indicating their potential involvement in regulating the tumor immune microenvironment in PCa. A more comprehensive understanding of this mechanism could be obtained by conducting an integrated analysis across multiple omics platforms. The current research focused on the role of IQGAP3 and KIF4A in the tumor development process; nonetheless, how these molecules affect the response to ICI treatment at the molecular level must be elucidated at a later stage. 

There were various limitations to this study. Firstly, we utilized the TGCA-PRAD and ICGC databases for data mining and analysis. Future research requires additional independent data validation and the evaluation of the precision of our prognostic model. Additionally, we plan in the future to further validate this model using specimens from our center and multiple centers, aiming for its early translation into clinical practice. Secondly, the risk factors are expressed at a low level in the scRNA-seq dataset, and with the maturity of scRNA-seq technology, more datasets can be analyzed in the future. Moreover, more in vivo and in vitro experiments are necessary to verify the role of these molecules in future research.

## 5. Conclusions and Future Work

To summarize, we identified five novel risk genes from TCGA-PRAD and developed a new-pattern prognostic model to predict the prognosis and immune cell infiltration of PCa patients, especially those with GS 9–10. Furthermore, we validated the model using ICGC and confirmed the expression levels of these signatures in PCa cells and patients through scRNA-seq datasets and in vitro experiments. These findings have important implications for the early diagnosis, risk stratification, and treatment decisions of PCa patients in clinical treatment.

## Figures and Tables

**Figure 1 diagnostics-13-01997-f001:**
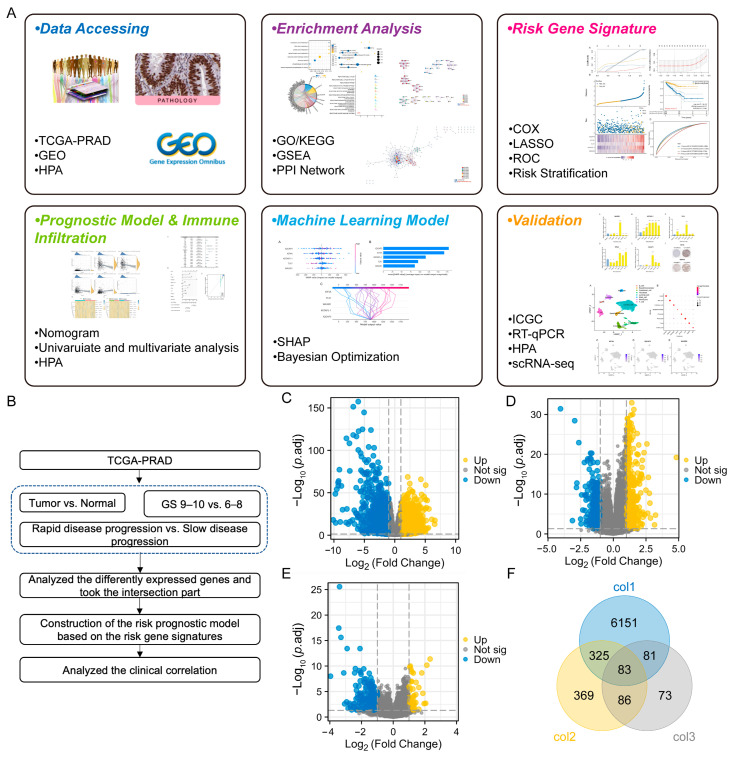
Screening of differentially expressed genes (DEGs) for the PCa risk prognostic model. (**A**,**B**) Workflow and study process overview. (**C**) Volcano plot showing DEGs in prostate cancer (PCa) compared to normal tissues. Col1 indicates strandedness for this cohort. (**D**) Volcano plot showing DEGs in PCa patients with GS 9–10 versus GS 6–8. Col2 indicates strandedness for this cohort. (**E**) Volcano plot showing DEGs in patients with faster-progression prostate cancer versus slower-progression prostate cancer. Col3 indicates strandedness for this cohort. (**F**) Venn diagram illustrating the overlap of DEGs among the three cohorts.

**Figure 2 diagnostics-13-01997-f002:**
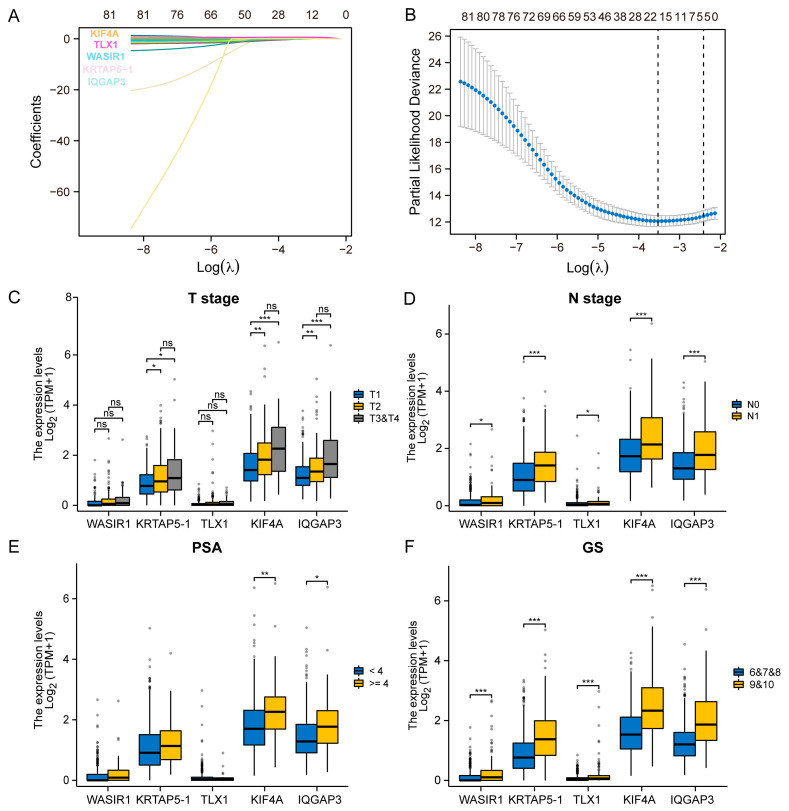
Identification of prognosis-related genes and association analysis with clinicopathological features. (**A**,**B**) LASSO regression plots for 83 genes in three cohorts. (**C**) Histogram of the expression level statistics of the prognosis-associated genes WASIR1, KRTAP5-1, TLX1, KIF4A, and IQGAP3 in T2, T3, and T4 patients with PCa. (**D**) Histogram of the expression level statistics of WASIR1, KRTAP5-1, TLX1, KIF4A, and IQGAP3 in N0 and N1 patients with PCa. (**E**) Histogram of the expression level statistics of WASIR1, KRTAP5-1, TLX1, KIF4A, and IQGAP3 in different PSA-level patients with PCa. (**F**) Histogram of the correlation analysis between the expression level of WASIR1, KRTAP5-1, TLX1, KIF4A, and IQGAP3 and GS in two groups. Significance levels are represented by asterisks: * *p* < 0.05, ** *p* < 0.01, *** *p* < 0.001, where * *p* is the reference level of significance.

**Figure 3 diagnostics-13-01997-f003:**
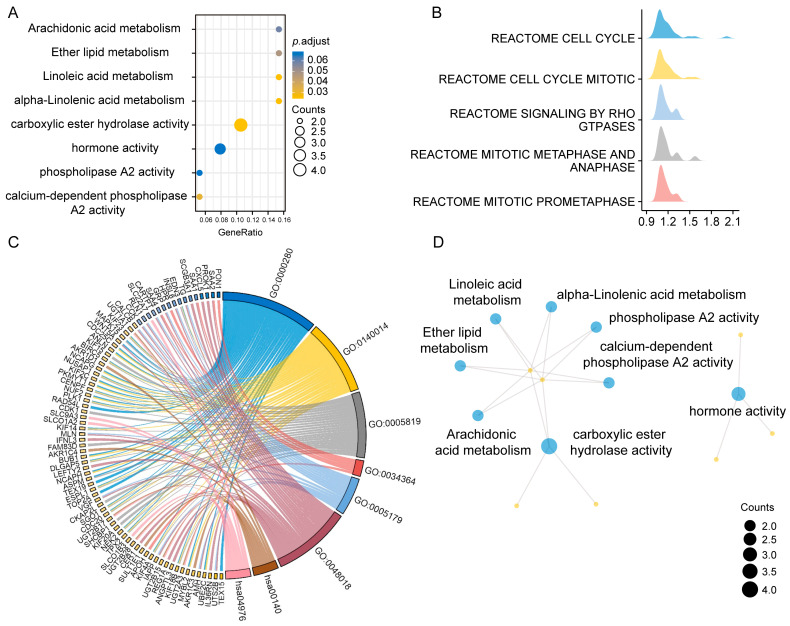
Results of the integration enrichment analysis of 83 genes common to the three cohorts and gene datasets for PCa patients with GS 9–10. (**A**) Bubble plot of possible functions of the 83 genes common to the three cohorts. (**B**) Graph of the results of GSEA (adjust *p* value < 0.05). (**C**) Chord diagram of the possible functions of differentially expressed genes in PCa patients with GS 9–10 based on GO/KEGG enrichment (adjust *p* value < 0.05). (**D**) Gene network visualization utilizing KEGG and GO enrichment analyses of 83 genes.

**Figure 4 diagnostics-13-01997-f004:**
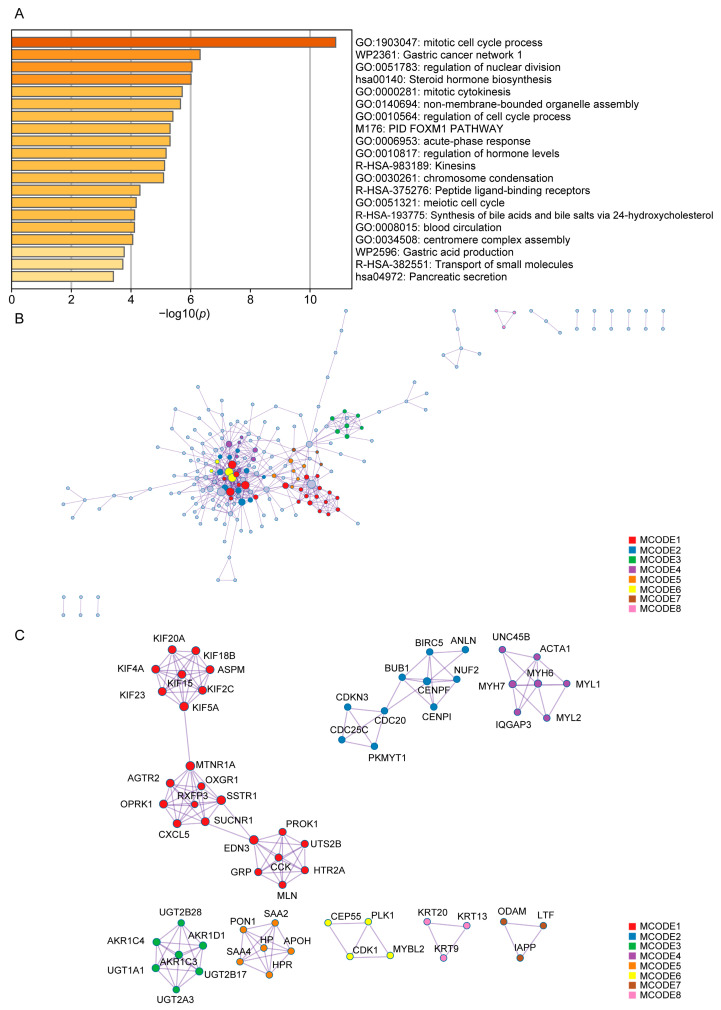
PPI network of the gene datasets for PCa patients with GS 9–10 from TCGA-PRAD. (**A**) Histogram of the enrichment results using Metascape network analysis (adjusted *p*-value < 0.05). (**B**) Schematic diagram of the PPI network for PCa patients with GS 9–10. (**C**) Schematic diagram of the hub gene for PCa patients with GS 9–10.

**Figure 5 diagnostics-13-01997-f005:**
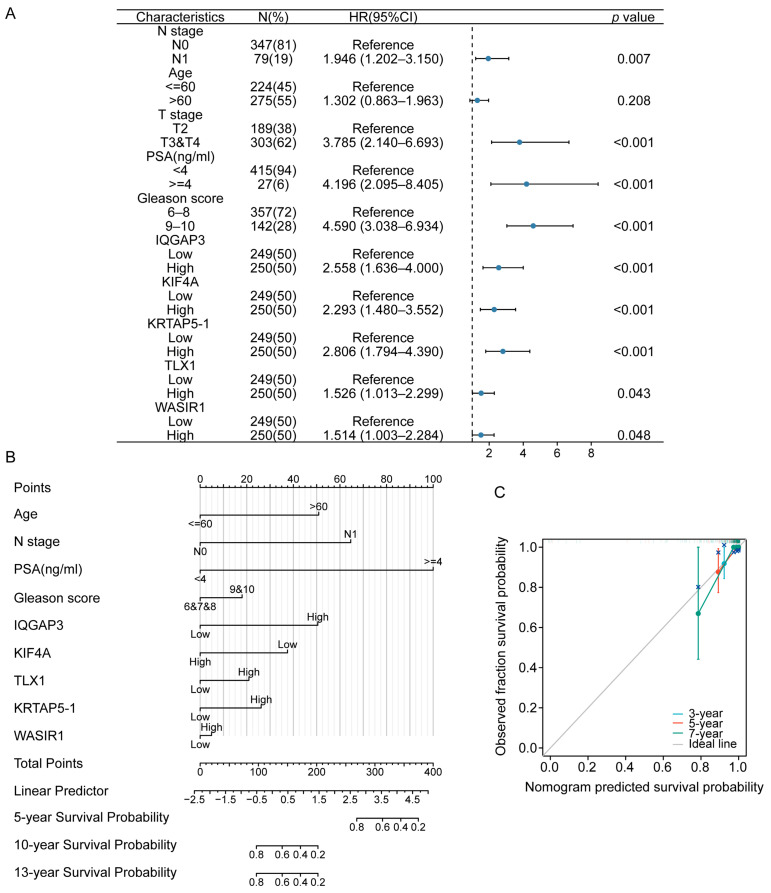
Univariable COX regression model and nomogram for prognostic prediction of PCa patients. (**A**) Forest plot showing hazard ratios derived from the univariable Cox regression model associated with progression-free interval (PFI). (**B**) Nomogram for the prediction of overall survival (OS) in PCa patients. (**C**) Calibration curve evaluating the accuracy of nomogram predictions.

**Figure 6 diagnostics-13-01997-f006:**
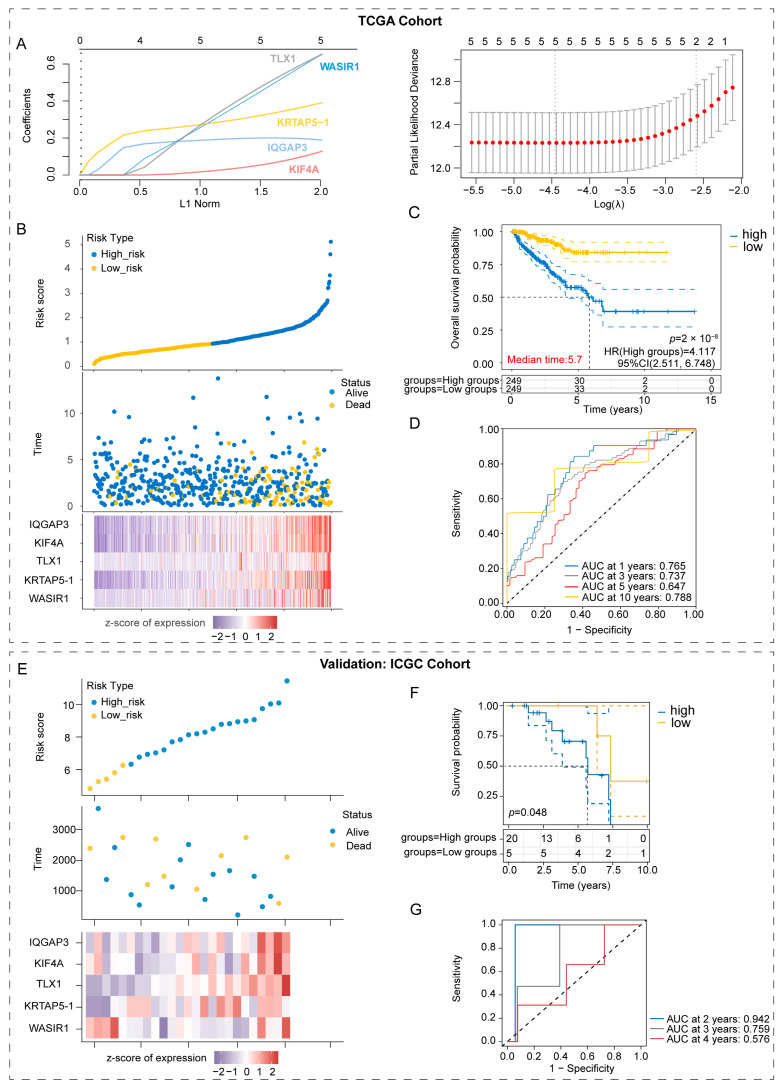
Prognostic model construction based on risk gene signatures for PCa patients. (**A**) LASSO regression plots displaying the top five genes. (**B**) Scatter plot of risk scores ranked from low to high, along with corresponding survival times and status distribution among different PCa samples. Heatmap depicting gene expression in the prognostic model. (**C**) Kaplan-Meier curves comparing high-risk patients to low-risk PCa patients in TCGA-PRAD. (**D**) Receiver operating characteristic (ROC) curves assessing the ability of this prognostic model to predict progression-free survival (PFS) at 1, 3, 5, and 10 years for PCa patients in TCGA-PRAD. (**E**) Scatter plot of risk scores ranked from low to high, along with corresponding survival times and status distribution among different PCa samples. Heatmap depicting gene expression in the prognostic model in ICGC. (**F**) Kaplan-Meier curves comparing high-risk patients to those with low-risk prostate cancer in the ICGC cohort. (**G**) ROC curves evaluate the prognostic model’s ability to predict PFS at 2, 3, and 4 years for PCa patients in ICGC.

**Figure 7 diagnostics-13-01997-f007:**
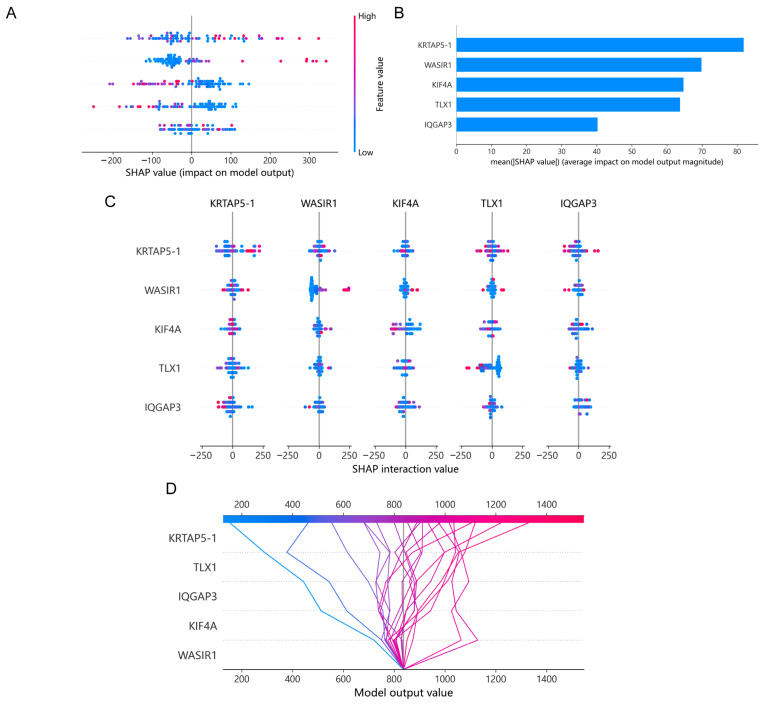
The importance and interaction of molecular expression features with regard to prognosis for PCa patients using a machine learning model. (**A**,**B**) Comparison of the importance of risk molecular expression level features with the SHAP estimation of randomly generated numbers. (**C**) Interactive plot. (**D**) Decision plot.

**Figure 8 diagnostics-13-01997-f008:**
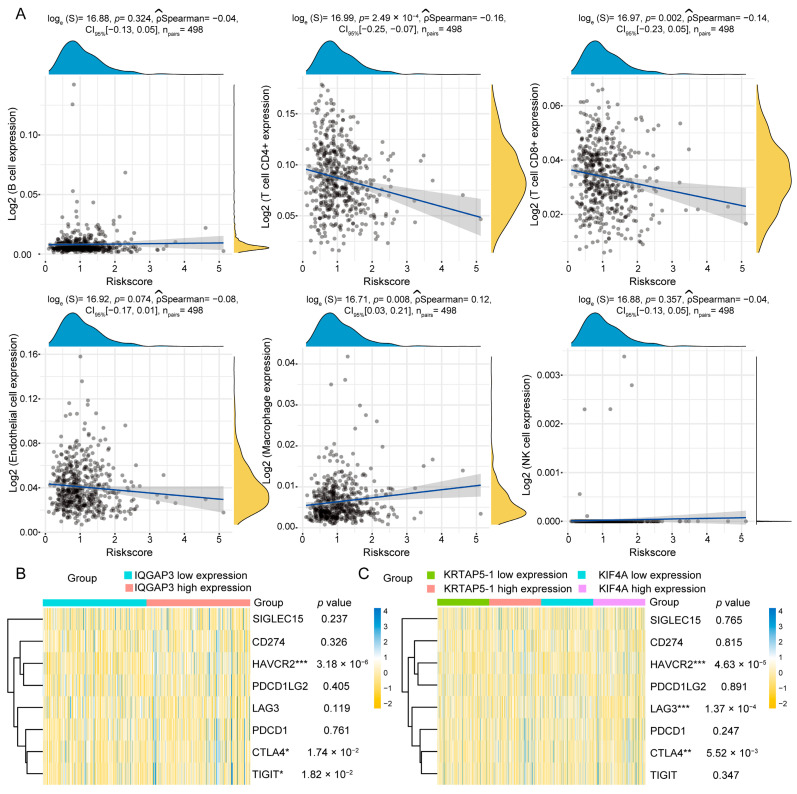
Correlation analysis of prognostic model and immunotherapy-associated markers. (**A**) Scatter plot depicting the relationship between the prognostic model and immune cell infiltration in prostate cancer tissues, as assessed by EPIC. (**B**) Heatmap illustrating the association between the IQGAP3 expression level and immune checkpoint-related genes. (**C**) Heatmap displaying the correlation between KRTAP5-1 and KIF4A expression levels and immune checkpoint-related genes. Significance levels are represented by asterisks: * *p* < 0.05, ** *p* < 0.01, *** *p* < 0.001, where * *p* is the reference level of significance.

**Figure 9 diagnostics-13-01997-f009:**
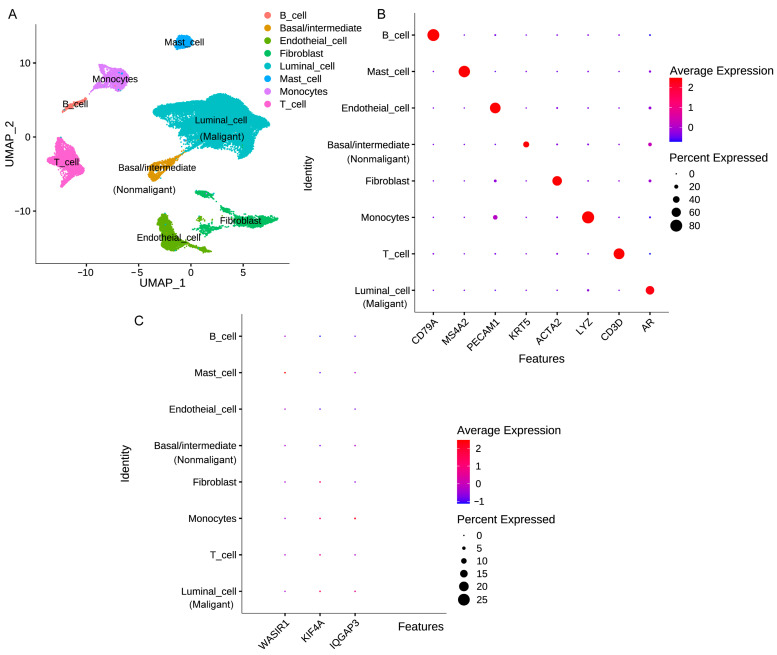
The expression levels of risk genes across different subpopulations of cells from GSE 141445. (**A**) Cell clustering results. (**B**) Expression patterns of different cell subgroups were identified through cell marker-based identification and clustering. (**C**) The dot plot of distribution for KIF4A, IQGAP3, and WASIR1 expression within subgroups of cells.

**Figure 10 diagnostics-13-01997-f010:**
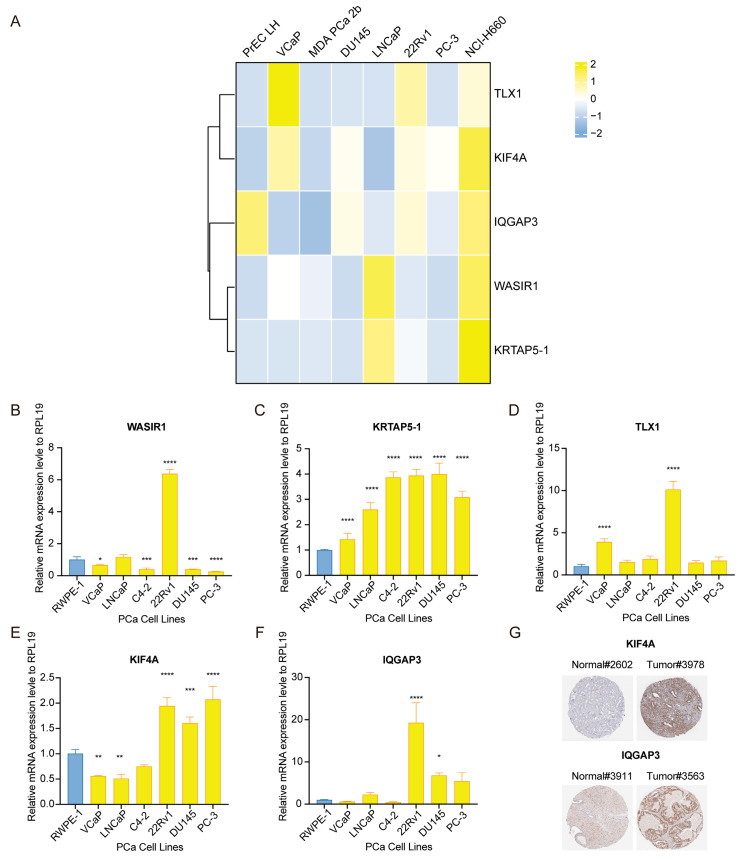
Confirmation of risk gene signatures in PCa cells and patients. (**A**) Heatmap showing the expression pattern of risk factors in CCLE. (**B**–**F**) Measurement of mRNA expression levels of WASIR1, KRTAP5-1, TLX1, KIF4A, and IQGAP3 via RT-qPCR in different PCa cell lines. (**G**) Assessment of KIF4A and IQGAP3 protein expression levels in normal and PCa tissues based on information from the HPA database. Significance levels are represented by asterisks: * *p* < 0.05, ** *p* < 0.01, *** *p* < 0.001, **** *p* < 0.0001, where * *p* is the reference level of significance.

## Data Availability

Our data were mined and analyzed using TCGA, GEO, and HPA databases. Further inquiries regarding the original data can be addressed to the corresponding author.

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
