# Peer review of "Crafting a Personalized Prognostic Model for Malignant Prostate Cancer Patients Using Risk Gene Signatures Discovered through TCGA-PRAD Mining, Machine Learning, and Single-Cell RNA-Sequencing"

_diagnostics, 2023, doi:10.3390/diagnostics13121997_

Round 1

Reviewer 1 Report

The study conducted by Lyu et al analyzed gene expression data from The Cancer Genome Atlas (TCGA) using multiple gene datasets to establish a risk model for predicting the overall survival (OS) of patients with prostate cancer (PCa). They utilized the Least Absolute Selection and Shrinkage Operator (LASSO) regression method and incorporated clinical variables into a nomogram. They also employed a machine learning model to examine the impact of risk factors on PCa prognosis and analyzed the relationship between their prognostic model and immune cell infiltration. Their findings were validated using GEO datasets, PCa cell lines, and tumor tissues from the Human Protein Atlas (HPA) database. The analysis identified 83 differentially expressed genes. After enrichment analysis, hormone metabolism and the cell cycle emerged as major drivers of malignant transformation in PCa. Further, WASIR1, 25 KRTAP5-1, TLX1, KIF4A, and IQGAP3 were determined as significant risk factors for OS and progression-free survival (PFS). I found the manuscript difficult to follow because many analyses lack clear motivation or rationalization. I have the following major concerns:

1.      The authors need to validate the 5 gene signature in an independent cohort.

2.      Figure 2A. It is unclear which lines correspond to the 5 selected genes and why exactly 5 genes were selected.

3.      What is the distribution of T stage, N stage, PSA and Gleason score across the data partitions used for differential expression analysis? If these values differ across partitions, then the gene expression may just be a spurious association.

4.      Figure 3. It is unclear how the enrichment analysis relates to the 5 genes. Are the 5 genes leading edge genes in any of the highlighted pathways? If the purpose is solely diagnostic, why perform enrichment analysis? If the purpose is understanding the mechanism of action, then why limit to 5 genes?

5.      Figure 4. The authors should explain in the text how the results are similar between Figures 3 and 4.

6.      What does Figure 4B show? The figure and legend lack any information to make sense of it.

7.      How do the 5 highlighted genes connect with data shown in Figure 4.

8.      Figure 5. It is unclear if gene expression improves diagnostic performance of the model. In the univariate model, Gleason score and PSA show greater hazard ratios. What is the point of including gene expression if the other variables perform better? If inclusion of gene expression does not significantly improve the model, what is the novelty in this paper?

9.      Figure 6. Was any cross-validation used here? Authors need to account for potential overfitting in their analyses.

10.   Figure 7. The outcomes here look very similar to the hazard model with TLX1 and WASIR showing lowest hazard ratios and importance scores. What additional information is gained in this analysis?

11.   I like the idea of integrating scRNAseq data. However, the current analysis is unclear. In the FeaturePlots (Fig9C, D, E) one cannot recognize any patterns. The authors need to use a better visualization. Potentially Dotplot like in Fig9B can be used here.

12.   Based on annotation in Fig9A it is not clear which cells are tumor cells and which ones are not. It is impossible to assess the results.

13.   Given that these 5 genes were differentially expressed at the bulk level the authors need to investigate which mechanism at the single cell level is more likely to explain the observation. A) Is there a change in frequency of cell types or B) are the genes differentially up/down regulated within each cell type?

14.   The authors should make use of DepMap resource for their cell line analysis (Figure 10).

15.   The authors need to provide their analysis code to ensure reproducibility (i.e. Github).

English language needs to be improved. Legend text is missing sufficient detail.

Author Response

Point-to-Point Response to Reviewer 1

Thank you very much for providing us with constructive comments on our manuscript. We sincerely appreciate your efforts in helping us to improve our work. We have thoroughly revised the manuscript according to your valuable feedback and have addressed all of your concerns. In addition, we have included new analyses to further strengthen our research. We hope that these adjustments have satisfactorily addressed any issues raised during the review process. All modified parts in the revised manuscript are displayed in red font. Due to the limitations of the system's text box, all modifications made to the images can be referenced in the attached file.

Revision 1: We have edited the manuscript for language by native speaker and made structural adjustments.

Revision 2: We have added background knowledge and identified existing issues in the introduction section, further highlighting the novelty of our research.

Revision 3: We have validated the risk model constructed in this study using an external data set, ICGC.

Revision 4: We have used an optimized machine learning model to add a validation set to assess the impact of risk factors on prognosis.

Revision 5: We have included CCLE data in the expression analysis of the validation factors.

Revision 6: In the discussion section, we have incorporated the latest research advances and provided a future outlook for this field.

Question 1: The authors need to validate the 5 gene signature in an independent cohort.

Response:

Thank you for your valuable comments. Your suggestions have greatly helped us improve our paper. We deeply regret that we did not consider this issue beforehand. The WASIR1 in our model is a lncRNA. We searched the prostate cancer dataset in GEO, but due to the lack of clinical information and encoding information of lncRNA, such as OS, PFS, etc., we were unable to verify the model.

Therefore, we used the data from the ICGC database and identified 25 patients who met the validation criteria and validated our 5-gene risk model. The results showed that in the ICGC cohort, the AUC value of the 5-gene risk model for predicting PFS at two years was 0.942, at three years was 0.759, and at four years was 0.576 (as shown in Figure 6 in revised manuscript page 11). However, because of the limited sample size, further validation studies with a larger sample size are needed in the future.

Revised Figure 6:

Figure 6. Prognostic model construction based on risk gene signatures for PCa patients. (A) LASSO regression plots displaying the top five genes. (B) Scatter plot of risk scores ranked from low to high, along with corresponding survival times and status distribution among different PCa samples. Heatmap depicting gene expression in the prognostic model. (C) Kaplan–Meier curves comparing high-risk patients to those with low-risk PCa patients in TCGA-PRAD. (D) Receiver operating characteristic (ROC) curves assessing the ability of this prognostic model to predict progression-free survival (PFS) at one, three, five, and ten years for PCa patients in TCGA-PRAD. (E) Risk scores and survival status chart of patients in the ICGC cohort. (F) Heatmap showing the expression of 5 risk factors in the ICGC cohort. (G) Kaplan–Meier curves comparing high-risk patients to those with low-risk prostate cancer in the ICGC cohort. (H) ROC curves evaluate the prognostic model's ability to predict PFS at 2, 4, and 4 years for PCa patients in ICGC.

Question 2: Figure 2A. It is unclear which lines correspond to the 5 selected genes and why exactly 5 genes were selected.

Response:

Thank you for your valuable feedback. Your comments have been extremely helpful in improving our manuscript. We selected 5 genes based on the following criteria:

(1) We removed unannotated non-coding molecules, such as AL645608.8 and AC010641.1, whose functions are currently unclear, because prostate cancer patients with Gleason score (GS) 9-10 have poor prognosis while overall survival is long.

(2) To simplify the model, we referred to recent literature where the number of risk factors was typically around 3-6. Therefore, in our study, we determined the number of risk genes to be 5, and ultimately selected WASIR1, KRTAP5-1, TLX1, KIF4A, and IQGAP3 to construct the risk model in our manuscript.

Regarding the issue of unclear gene labeling in Figure 2A, we reanalyzed the data and labeled the 5 risk genes in Figure 2 (page 7 in revised manuscript).

Revised Figure 2A:

Figure 2. Identification of prognosis-related genes and association analysis with clinicopathological features. (A) LASSO regression plots for 83 genes in three cohorts.

References:

Yang P, Lu J, Zhang P, Zhang S. Comprehensive Analysis of Prognosis and Immune Landscapes Based on Lipid-Metabolism- and Ferroptosis-Associated Signature in Uterine Corpus Endometrial Carcinoma. Diagnostics (Basel). 2023;13(5):870. Published 2023 Feb 24. doi:10.3390/diagnostics13050870

Wu Z, Lu Z, Li L, et al. Identification and Validation of Ferroptosis-Related LncRNA Signatures as a Novel Prognostic Model for Colon Cancer. Front Immunol. 2022;12:783362. Published 2022 Jan 26. doi:10.3389/fimmu.2021.783362

Question 3: What is the distribution of T stage, N stage, PSA and Gleason score across the data partitions used for differential expression analysis? If these values differ across partitions, then the gene expression may just be a spurious association.

Response:

Thank you for your valuable input. Your comments are crucial to the success of our manuscript. We acknowledge that different clinical variables may have different distributions among the groups, including both normal and non-normal distributions. Hence, we chose appropriate statistical methods for each data type to avoid false associations.

For sample data consisting of two groups, we used t-test with equal variance or Welch's t-test with unequal variance if the data followed a normal distribution. If the data was non-normal, we applied Wilcoxon rank-sum test. For data composed of three or more groups, with normality and unequal variances, we adopted the One-way ANOVA approach and Tukey HSD post hoc test for multiple comparisons to reduce false associations.

To address your comments, we combined the analyses of T3 and T4 prostate cancer. Furthermore, to minimize the sequencing errors and discrepancies between different samples, we normalized the TPM values extracted from the sequencing data, which helped us in controlling the data quality during analysis.

Question 4: Figure 3. It is unclear how the enrichment analysis relates to the 5 genes. Are the 5 genes leading edge genes in any of the highlighted pathways? If the purpose is solely diagnostic, why perform enrichment analysis? If the purpose is understanding the mechanism of action, then why limit to 5 genes?

Response:

Thank you for your valuable feedback. Your comments have been very helpful for our paper. Due to the limited number of genes involved (only five), we didn't conduct functional enrichment analysis. The current enrichment analysis methods such as GO/KEGG and GSEA require a larger number of genes (more than ten or even hundreds) to produce more accurate results. Therefore, we constructed a PPI network based on gene expression information from prostate cancer patients with GS9-10 in Figure 4. The results shows that KIF4A and IQGAP3 (page 9 in revised manuscript) family molecules were defined as hub genes, suggesting their important role in malignant transformation of GS 9-10 prostate cancer, and highlighting the key molecules that we selected.

Our study was based on the expression profiles of GS 9-10 prostate cancer because these patients have a high degree of malignancy and resistance to various treatments (such as endocrine therapy, radiotherapy, and chemotherapy). However, there is a lack of research on GS 9-10 cells and animal models. Therefore, we used enrichment analysis to explore the possible mechanism of malignant transformation of GS 9-10 and screened out key molecules for early identification of highly malignant prostate cancer. Our study not only explores the possible biological behaviors and malignant transformation mechanisms of GS 9-10 prostate cancer but also provides risk factors for model construction by screening out five relevant markers. If there are too many molecules, there will be great pressure on the subsequent validation and transformation studies. Therefore, we chose to screen out only five molecules.

References:

Subramanian A, Tamayo P, Mootha VK, et al. Gene set enrichment analysis: a knowledge-based approach for interpreting genome-wide expression profiles. Proc Natl Acad Sci U S A. 2005;102(43):15545-15550. doi:10.1073/pnas.0506580102

Mootha VK, Lindgren CM, Eriksson KF, et al. PGC-1alpha-responsive genes involved in oxidative phosphorylation are coordinately downregulated in human diabetes. Nat Genet. 2003;34(3):267-273. doi:10.1038/ng1180

Question 5: Figure 4. The authors should explain in the text how the results are similar between Figures 3 and 4.

Response:

Thank you for your valuable feedback. Your comments have been very helpful for our paper. We apologize for not being clear enough in the manuscript. Both Figure 3 and Figure 4 are results of enrichment analysis. The results of the enrichment analysis show that biological behaviors such as cell cycle regulation and androgen synthesis may be key events in prostate cancer patients with GS 9-10. We have clarified this in the revised manuscript (lines 255 to 258), and Figure 4 is a further supplement to Figure 3. Thank you again for your valuable feedback, which has helped us improve the quality of our paper.

Question 6: What does Figure 4B show? The figure and legend lack any information to make sense of it.

Response:

Thank you for your valuable feedback on our manuscript. We appreciate your helpful comments. Figure 4B is a classic protein-protein interaction (PPI) network diagram, which represents the interactions between proteins and consists of multiple nodes and edges. Nodes represent proteins, while edges represent their interactions. By analyzing the relationships between the nodes and edges, we can understand the patterns of protein interactions and network topology structures, thereby identifying important proteins, pathways, modules, or mechanisms.

PPI network analysis can help to identify potential cancer biomarkers and therapeutic targets. By analyzing the topology structure of PPI networks, we can gain insights into the relationship between proteins and reveal the complex regulatory mechanisms of biological systems. As shown in Figure 4C, KIF4A and IQGAP3 may be key regulatory nodes in GS 9-10 prostate cancer and are potential molecular targets for further investigation.

Drawing PPI networks is a commonly used method of analysis, and examples from the literature are provided below.

Additionally, we have revised the figure caption to improve clarity as follows (lines 316 to 319 in revised manuscript): Figure 4 depicts the PPI network of gene datasets for PCa patients with GS 9-10 from TCGA-PRAD. (A) shows a histogram of enrichment results using Metascape network analysis (adjusted p value < 0.05). (B) is a schematic diagram of the PPI network for PCa patients with GS 9-10. (C) is a schematic diagram of the hub gene for PCa patients with GS 9-10.

Once again, thank you for your valuable suggestions.

References:

Liu M, Dou Y, Sun R, Zhang Y, Liu Y. Molecular mechanisms for alcoholic hepatitis based on analysis of gene expression profile. Hepat Mon. 2015;15(5):e27336. Published 2015 May 23. doi:10.5812/hepatmon.15(5)2015.27336

Question 7: How do the 5 highlighted genes connect with data shown in Figure 4.

Response:

Thank you for your valuable feedback, which has been tremendously helpful in improving the quality of our manuscript. We would like to clarify that the five genes were excavated from prostate cancer patients with GS 9-10. Figure 4 analyzes the potential biological behavior of highly malignant prostate cancer patients and identifies key molecules through MCODE analysis. As indicated by the results in Figure 4C, KIF4A and IQGAP3 were defined as critical genes, and we have added relevant content to the manuscript to suggest their potential important role in the malignant biological behavior of prostate cancer patients with GS 9-10. These two genes could serve as crucial targets for future experimental verification studies. Other molecules were not defined as hub genes, which might be attributed to their noninvolvement in critical nodes.

Figure 4C:

Figure 4. (C) Schematic diagram of the hub gene for PCa patients with GS 9-10.

Question 8: Figure 5. It is unclear if gene expression improves diagnostic performance of the model. In the univariate model, Gleason score and PSA show greater hazard ratios. What is the point of including gene expression if the other variables perform better? If inclusion of gene expression does not significantly improve the model, what is the novelty in this paper?

Response:

Thank you very much for your valuable feedback on our manuscript. Your comments have been tremendously helpful to us. We would like to address the issue raised in your comment regarding the necessity of including gene expression data in our analysis.

Currently, the most used clinical nomogram only contains clinical data, such as the MSKCC nomogram (PSA and Date). However, we believe that it is necessary to incorporate gene expression data into our analysis, as it is a trend in disease risk stratification research and provides rich biological information, such as biological signaling pathways and regulatory networks (Figures 3,4), which cannot be provided by traditional clinical indicators. Moreover, even if gene expression data do not directly improve the predictive performance of the model, they still provide new insights and breakthroughs for future research and clinical applications, such as new biomarkers and targeted therapeutic drugs.

In this study, we chose to screen for molecules based on the molecular expression profiles of patients with clinically most aggressive GS 9-10 prostate cancer, rather than selecting molecules from the regulation of microscopic cellular programmed cell death. We believe that the screened molecules targeting GS 9-10 can reflect the overall malignancy of the disease. These screened molecules are novel and previously unreported, and our study has combined advanced machine learning models and single-cell RNA sequencing technologies, making it innovative. The construction of a new predictive model is one of the main achievements of our research.

Thank you for your suggestion regarding the comparison of diagnostic efficacy with traditional tools. We apologize for not performing such an evaluation in our study owing to the limited clinical information available from TCGA-PRAD and the relatively short follow-up duration. To address this issue, we plan to conduct a series of clinical studies in the future to evaluate the performance of our model and to facilitate its clinical translation.

References:

Cucchiara V, Cooperberg MR, Dall'Era M, et al. Genomic Markers in Prostate Cancer Decision Making. Eur Urol. 2018;73(4):572-582. doi:10.1016/j.eururo.2017.10.036

Prostate Cancer Nomograms: PSA Doubling Time | Memorial Sloan Kettering Cancer Center (mskcc.org)

Question 9: Was any cross-validation used here? Authors need to account for potential overfitting in their analyses.

Response:

Thank you for your valuable feedback. We have reanalyzed the data as suggested. We partitioned TCGA-PRAD patients into training, testing and validation sets in a 6:2:2 ratio. We then utilized machine learning models, including nonlinear regression, random forest, and XGBoost to examine the impact of risk factors on the progression-free interval (PFI) of PCa patients. To optimize model hyperparameters such as maximal number of samples in the leaf and tree depth, we employed RandomizedSearchCV and GridSearchCV from the Python scikit-learn library with 5-fold cross-validation and Bayesian optimization. Additionally, we examined feature importance and interactions using SHAP (Python, version 3.9.2). The figure was shown in Figure 7. We have found that KRTAP5-1 and WASIR1 are the most important factors for prognosis in the output model, which differs from our previous results. We believe this may be due to the increased validation set and the use of Bayesian optimization.

Question 10: Figure 7. The outcomes here look very similar to the hazard model with TLX1 and WASIR showing lowest hazard ratios and importance scores. What additional information is gained in this analysis?

Response:

Thank you very much for your valuable feedback on our paper. Your comments have been incredibly helpful in improving the quality of our research. As per your suggestion, we have performed a rigorous reanalysis using an optimized machine learning model, the detailed methodology of which is explained in response to Question 9. Machine learning algorithms are capable of adaptively adjusting their behavior based on different data inputs to achieve the best results, while also performing complex inference and prediction tasks. These algorithms can identify complex patterns within the data and use them for inference and prediction, helping us to predict future trends or behaviors and make corresponding decisions.

Therefore, the results shown in Figure 7 represent the output of the machine learning model, which demonstrate that WASIR1 and KRTAP5-1 are the most impactful progression-related genes in prostate cancer PFS, with the LASSO regression model also arriving at the same conclusion regarding WASIR1. Furthermore, Figure 7 uses an optimized machine learning model to evaluate the impact of risk gene expression on prognosis, providing additional and complementary analysis to the earlier results with an ability to analyze interaction effects among variables.

Revised Figure 7:

Figure 7. The importance and interaction of molecular expression feature on prognosis for PCa patients using a machine learning model. (A-B) Comparison of the importance of risk molecular expression level features with the SHAP estimation of randomly generated numbers. (C) Interactive plot. (D) Decision plot.

References:

Deo RC. Machine Learning in Medicine. Circulation. 2015;132(20):1920-1930. doi:10.1161/CIRCULATIONAHA.115.001593

MacEachern SJ, Forkert ND. Machine learning for precision medicine. Genome. 2021;64(4):416-425. doi:10.1139/gen-2020-0131

Question 11: I like the idea of integrating scRNAseq data. However, the current analysis is unclear. In the FeaturePlots (Fig9C, D, E) one cannot recognize any patterns. The authors need to use a better visualization. Potentially Dotplot like in Fig9B can be used here.

Response:

Thank you very much for your valuable feedback on our manuscript. Your comments have been very helpful for us in improving the quality of our work. Based on your suggestions, we have revised our manuscript by changing the original figure to a dot plot as shown in Figure 9C (page 14 in revised manuscript).

Revised Figure 9C:

Figure 9. The expression levels of risk genes across different subpopulations of cells from GSE 141445. (A) Cell clustering results. (B) Expression patterns of different cell subgroups were identified through cell marker-based identification and clustering. (C) The dot plot of distribution for KIF4A, IQGAP3, and WASIR1 expression within subgroups of cells.

Question 12: Based on annotation in Fig9A it is not clear which cells are tumor cells and which ones are not. It is impossible to assess the results.

Response:

Thank you very much for your feedback on our manuscript. Your comments have been extremely helpful in improving the quality of our work.

As per your suggestion, we would like to clarify that our single-cell RNA-seq dataset was constructed based on the GSE141445 dataset and molecular profiling was performed according to marker gene annotation and original literature analysis as well as clustering results. We have labeled the luminal clusters as tumor cells, which are malignant in nature, while the basal/intermediate cell clusters have been labeled as nonmalignant cells. We have included relevant annotations on the figures to help readers better understand the status of individual cell clusters shown in Figure 9. Specifically, we have updated the figure and made the corresponding modifications in Question 11 to improve the clarity and consistency of our presentation.

References:

Chen S, Zhu G, Yang Y, et al. Single-cell analysis reveals transcriptomic remodellings in distinct cell types that contribute to human prostate cancer progression. Nat Cell Biol. 2021;23(1):87-98. doi:10.1038/s41556-020-00613-6

Question 13: Given that these 5 genes were differentially expressed at the bulk level the authors need to investigate which mechanism at the single cell level is more likely to explain the observation. A) Is there a change in frequency of cell types or B) are the genes differentially up/down regulated within each cell type?

Response:

Thank you very much for your valuable feedback on our manuscript. Your comments have provided us with inspiring insights into the interpretation of our results and brought new ideas into our discussion of different outcomes. As prostate cancer is a hormone-dependent tumor, its cell lines can be classified as either hormone-sensitive or hormone-resistant, and according to the expression of AR as AR-expressing or non-AR-expressing cells. 22Rv1 is a castration-resistant prostate cancer cell line that expresses both AR and AR variants, which corresponds to the molecular expression characteristics of clinical prostate cancer patients. Therefore, these five risk factors were upregulated compared to normal prostatic epithelium in this cell line, and their expression was different from that of other cell lines.

Furthermore, we propose that hormones and AR may account for the differential expression patterns observed (lines 394 to 396 in revised manuscript).

References:

Namekawa T, Ikeda K, Horie-Inoue K, Inoue S. Application of Prostate Cancer Models for Preclinical Study: Advantages and Limitations of Cell Lines, Patient-Derived Xenografts, and Three-Dimensional Culture of Patient-Derived Cells. Cells. 2019;8(1):74. Published 2019 Jan 20. doi:10.3390/cells8010074

Question 14: The authors should make use of DepMap resource for their cell line analysis (Figure 10).

Response:

Thank you very much for your valuable feedback. Your comments have greatly helped us to improve our manuscript and enriched our data. In response to your suggestions, we have updated Figure 10 (page 15 in revised manuscript) by incorporating data from CCLE in DepMap. We analyzed the expression profile of the five risk factors in one normal prostatic epithelial cell line (PrEC LH) and seven prostate cancer cell lines (VCaP, MDA PCa 2b, DU145, LNCaP, 22Rv1, PC-3, NCI-H660) and presented the results using a heatmap. Interestingly, our study revealed a significant upregulation of these five risk factors in the neuroendocrine differentiated prostate cancer cell line (NCI-H660), indicating their potential correlation with malignancy and neuroendocrine differentiation in prostate cancer. Further investigations on this finding are warranted.

Revised Figure 10A:

Figure 10. Confirmation of risk gene signatures in PCa cells and patients. (A) Heatmap showing the expression pattern of risk factors in CCLE.

Question 15: The authors need to provide their analysis code to ensure reproducibility (i.e. Github).

Response:

Thank you very much for your valuable feedback. Your comments have greatly helped us to improve our manuscript. Unfortunately, due to the network VPN restriction, we were unable to upload our code to Github. However, we have uploaded the code to a cloud storage platform, Nutstore: https://www.jianguoyun.com/p/DQ_6CQwQn8LSCxjKzYYFIAA.

Thank you for your valuable feedback. We have taken your suggestions into consideration and worked with language editing services provided by the MDPI publishing house to improve the quality of our manuscript.

Certificate:

We greatly appreciate your efforts in reviewing our work and understand the importance of constructive criticism in enhancing the quality of our research. Your feedback has been instrumental in guiding us towards the best possible outcome.

Thank you once again for your time and expertise. It has been a pleasure working with you.

Reviewer 2 Report

Thank you for the opportunity to review this high quality work. I do not detect any issues with the work and endorse it for publication in its current form.

Author Response

Point-to-Point Response to Reviewer 2

Thank you very much for providing us with constructive comments on our manuscript. We sincerely appreciate your efforts in helping us to improve our work. We have thoroughly revised the manuscript according to your valuable feedback and have addressed all of your concerns. In addition, we have included new analyses to further strengthen our research. We hope that these adjustments have satisfactorily addressed any issues raised during the review process. All modified parts in the revised manuscript are displayed in red font.

Revision 1: We have edited the manuscript for language by native speaker and made structural adjustments.

Revision 2: We have added background knowledge and identified existing issues in the introduction section, further highlighting the novelty of our research.

Revision 3: We have validated the risk model constructed in this study using an external data set, ICGC.

Revision 4: We have used an optimized machine learning model to add a validation set to assess the impact of risk factors on prognosis.

Revision 5: We have included CCLE data in the expression analysis of the validation factors.

Revision 6: In the discussion section, we have incorporated the latest research advances and provided a future outlook for this field.

Thank you for your valuable feedback. We have taken your suggestions into consideration and worked with language editing services provided by the MDPI publishing house to improve the quality of our manuscript.

We greatly appreciate your efforts in reviewing our work and understand the importance of constructive criticism in enhancing the quality of our research. Your feedback has been instrumental in guiding us towards the best possible outcome.

Thank you once again for your time and expertise. It has been a pleasure working with you.

Reviewer 3 Report

1. How did you select and quality-control the TCGA-PRAD dataset that consists of different patient cohorts?

2. How generalizable and applicable is your prognostic model to other datasets or prostate cancer subtypes?

3. What are the roles and mechanisms of the risk gene signatures that you discovered in driving the malignancy of prostate cancer?

Fine

Author Response

Point-to-Point Response to Reviewer 3

Thank you very much for providing us with constructive comments on our manuscript. We sincerely appreciate your efforts in helping us to improve our work. We have thoroughly revised the manuscript according to your valuable feedback and have addressed all of your concerns. In addition, we have included new analyses to further strengthen our research. We hope that these adjustments have satisfactorily addressed any issues raised during the review process. All modified parts in the revised manuscript are displayed in red font. Due to the limitations of the system's text box, all modifications made to the images can be referenced in the attached file.

Revision 1: We have edited the manuscript for language by native speaker and made structural adjustments.

Revision 2: We have added background knowledge and identified existing issues in the introduction section, further highlighting the novelty of our research.

Revision 3: We have validated the risk model constructed in this study using an external data set, ICGC.

Revision 4: We have used an optimized machine learning model to add a validation set to assess the impact of risk factors on prognosis.

Revision 5: We have included CCLE data in the expression analysis of the validation factors.

Revision 6: In the discussion section, we have incorporated the latest research advances and provided a future outlook for this field.

Question 1: How did you select and quality-control the TCGA-PRAD dataset that consists of different patient cohorts?

Response:

Thank you for your valuable feedback. Your comments have been extremely helpful in improving our manuscript. Regarding TCGA-PRAD data, we selected patients with complete gene expression and clinical information for inclusion in the analysis, excluding those with missing values. All included patients are suitable for subsequent analysis. To avoid analysis errors among patients from different cohorts, corresponding statistical methods were used as appropriate. The specific methods used are as follows:

For sample data consisting of two groups, we used t-test with equal variance or Welch's t-test with unequal variance if the data followed a normal distribution. If the data was non-normal, we applied Wilcoxon rank-sum test. For data composed of three or more groups, with normality and unequal variances, we adopted the One-way ANOVA approach and Tukey HSD post hoc test for multiple comparisons to reduce false associations.

We combined the analyses of T3 and T4 prostate cancer to avoid spurious association. Furthermore, to minimize the sequencing errors and discrepancies between different samples, we normalized the TPM values extracted from the sequencing data, which helped us in controlling the data quality during analysis.

And We selected 5 genes based on the following criteria:

(1) We removed unannotated non-coding molecules, such as AL645608.8 and AC010641.1, whose functions are currently unclear, because prostate cancer patients with Gleason score (GS) 9-10 have poor prognosis while overall survival is long.

(2) To simplify the model, we referred to recent literature where the number of risk factors was typically around 3-6. Therefore, in our study, we determined the number of risk genes to be 5, and ultimately selected WASIR1, KRTAP5-1, TLX1, KIF4A, and IQGAP3 to construct the risk model in our manuscript.

We acknowledge that different clinical variables may have different distributions among the groups, including both normal and non-normal distributions. Hence, we chose appropriate statistical methods for each data type to avoid false associations.

References:

Yang P, Lu J, Zhang P, Zhang S. Comprehensive Analysis of Prognosis and Immune Landscapes Based on Lipid-Metabolism- and Ferroptosis-Associated Signature in Uterine Corpus Endometrial Carcinoma. Diagnostics (Basel). 2023;13(5):870. Published 2023 Feb 24. doi:10.3390/diagnostics13050870

Wu Z, Lu Z, Li L, et al. Identification and Validation of Ferroptosis-Related LncRNA Signatures as a Novel Prognostic Model for Colon Cancer. Front Immunol. 2022;12:783362. Published 2022 Jan 26. doi:10.3389/fimmu.2021.783362

Question 2: How generalizable and applicable is your prognostic model to other datasets or prostate cancer subtypes?

Response:

Thank you for your valuable comments. Your suggestions have greatly helped us improve our paper. We deeply regret that we did not consider this issue beforehand. The WASIR1 in our model is a lncRNA. We searched the prostate cancer dataset in GEO, but due to the lack of clinical information and encoding information of lncRNA, such as OS, PFS, etc., we were unable to verify the model.

Therefore, we used the data from the ICGC database and identified 25 patients who met the validation criteria and validated our 5-gene risk model. The results showed that in the ICGC cohort, the AUC value of the 5-gene risk model for predicting PFS at two years was 0.942, at three years was 0.759, and at four years was 0.576 (page 11 in revised manuscript). However, because of the limited sample size, further validation studies with a larger sample size are needed in the future.

Revised Figure 6:

Figure 6. Prognostic model construction based on risk gene signatures for PCa patients. (A) LASSO regression plots displaying the top five genes. (B) Scatter plot of risk scores ranked from low to high, along with corresponding survival times and status distribution among different PCa samples. Heatmap depicting gene expression in the prognostic model. (C) Kaplan–Meier curves comparing high-risk patients to those with low-risk PCa patients in TCGA-PRAD. (D) Receiver operating characteristic (ROC) curves assessing the ability of this prognostic model to predict progression-free survival (PFS) at one, three, five, and ten years for PCa patients in TCGA-PRAD. (E) Risk scores and survival status chart of patients in the ICGC cohort. (F) Heatmap showing the expression of 5 risk factors in the ICGC cohort. (G) Kaplan–Meier curves comparing high-risk patients to those with low-risk prostate cancer in the ICGC cohort. (H) ROC curves evaluate the prognostic model's ability to predict PFS at 2, 4, and 4 years for PCa patients in ICGC.

Regarding other subtypes of prostate cancer, such as neuroendocrine prostate cancer (NEPC), there is currently a lack of datasets on these specific subtypes. However, as shown in Figure 10A based on the CCLE dataset, these risk molecules are significantly upregulated in NEPC cells (NCI-H660), suggesting that they may play an important role in NEPC differentiation and malignant biology. Therefore, we plan to design corresponding in vitro and in vivo experiments to validate this possibility. We appreciate your valuable suggestion for future directions of our study.

Question 3: What are the roles and mechanisms of the risk gene signatures that you discovered in driving the malignancy of prostate cancer?

Response:

Thank you for your inquiry. Your questions are valuable and constructive to our ongoing research. Through a combined screening of gene expression profiles in highly malignant prostate cancer patients from the TCGA-PRAD database, we have identified WASIR1, KRTAP5-1, TLX1, KIF4A, and IQGAP3 as important risk factors. However, due to the current requirements of functional enrichment analysis that require the inclusion of many molecules (hundreds to thousands), the small number of five molecules in our study prevented us from conducting functional enrichment analysis. We conducted a literature search on these molecules in prostate cancer and summarized the findings as follows:

Molecules

Diseases

Research Progress

WASIR1

-

WASIR1 (WASH And IL9R Antisense RNA 1) is an RNA gene that belongs to the lncRNA class. There have been few studies on this lncRNA so far.

KRTAP5-1

-

In the hair cortex, hair keratin intermediate filaments are embedded in an interfilamentous matrix, consisting of hair keratin-associated protein (KRTAP), which are essential for the formation of a rigid and resistant hair shaft through their extensive disulfide bond cross-linking with abundant cysteine residues of hair keratins. The matrix proteins include the high-sulfur and high-glycine-tyrosine keratins. KRTAP5-1 is related to protein binding. There is no relevant research on KRTAP5-1 in prostate cancer.

TLX1

Acute T-lymphocytic leukemia and lymphoma

Current studies on TLX1 mainly focus on the hematopoietic system, which can induce the development of hematological malignancies.

KIF4A

Esophageal squamous carcinoma

KIF4A promotes proliferation and migration of esophageal squamous cell carcinoma cells via the Hippo pathway.

Cholangiocarcinoma

KIF4A is a biomarker for biliary tract carcinoma.

Pancreatic ductal adenocarcinoma (PDAC)

KIF4A promotes proliferation and invasion of PDAC.

Prostate cancer

Increased KIF4A expression may potentially predict poor BCR-free survival in PCa patients.

KIF4A forms a positive feedback regulatory loop with AR, promoting resistance of cells to androgen.

IQGAP3

Prostate cancer

IQGAP3 is a prognostic biomarker for prostate cancer that is correlated with the expression of PLK1 and TOP2A, although the mechanism is not clear.

Bladder cancer

Cell-free DNA of IQGAP3 in urine is a biomarker for non-muscle-invasive bladder cancer.

Therefore, IQGAP3 and KIF4A have related studies in prostate cancer, which further supports our selection of these molecules as risk factors in the model that we engineered. However, there is limited research on WASIR1, KRTAP5-1, and TLX1 in prostate cancer, and further in vivo and in vitro experiments are needed to investigate their functions.

Thank you for your valuable feedback. We have taken your suggestions into consideration and worked with language editing services provided by the MDPI publishing house to improve the quality of our manuscript.

We greatly appreciate your efforts in reviewing our work and understand the importance of constructive criticism in enhancing the quality of our research. Your feedback has been instrumental in guiding us towards the best possible outcome.

Thank you once again for your time and expertise. It has been a pleasure working with you.

References

Riz I, Hawley TS, Luu TV, Lee NH, Hawley RG. TLX1 and NOTCH coregulate transcription in T cell acute lymphoblastic leukemia cells. Mol Cancer. 2010;9:181. Published 2010 Jul 9. doi:10.1186/1476-4598-9-181

Riz I, Akimov SS, Eaker SS, et al. TLX1/HOX11-induced hematopoietic differentiation blockade. Oncogene. 2007;26(28):4115-4123. doi:10.1038/sj.onc.1210185

De Keersmaecker K, Ferrando AA. TLX1-induced T-cell acute lymphoblastic leukemia. Clin Cancer Res. 2011;17(20):6381-6386. doi:10.1158/1078-0432.CCR-10-3037

Zhang DY, Ma SS, Sun WL, Lv XCH, Lu Z. KIF4A as a novel prognostic biomarker in cholangiocarcinoma. Medicine (Baltimore). 2021;100(21):e26130. doi:10.1097/MD.0000000000026130

Sun X, Chen P, Chen X, et al. KIF4A enhanced cell proliferation and migration via Hippo signaling and predicted a poor prognosis in esophageal squamous cell carcinoma. Thorac Cancer. 2021;12(4):512-524. doi:10.1111/1759-7714.13787

Gao H, Chen X, Cai Q, Shang Z, Niu Y. Increased KIF4A expression is a potential prognostic factor in prostate cancer. Oncol Lett. 2018;15(5):7941-7947. doi:10.3892/ol.2018.8322

Xu Y, Kim YH, Jeong P, et al. Urinary Cell-Free DNA IQGAP3/BMP4 Ratio as a Prognostic Marker for Non-Muscle-Invasive Bladder Cancer. Clin Genitourin Cancer. 2019;17(3):e704-e711. doi:10.1016/j.clgc.2019.04.001

Mei W, Dong Y, Gu Y, et al. IQGAP3 is relevant to prostate cancer: A detailed presentation of potential pathomechanisms [published online ahead of print, 2023 Jan 18]. J Adv Res. 2023;S2090-1232(23)00028-0. doi:10.1016/j.jare.2023.01.015

Reviewer 4 Report

- Paper is well written. Authors should add a little background of the study and limitations of the existing works.  

- Please improve the overall readability of the paper.

- Numerous solutions to this problem have been proposed in past research. Please explain the novelty and contribution of the study. Why is this study important and what academic values it adds to the field?

- Some Paragraphs in the paper can be merged and some long paragraphs can be split into two.

- The paper is a bit too long; the author should try to present the key points to show their contributions while some basic definitions can be shortened to some extent.

- What are the limitations of this study?

- Authors should summarize the recent works in the form of a table.

- The contributions of the paper may be highlighted as bullets in the Introduction. 

- Authors should discuss and add the below reference in the introduction:

* Clinical theragnostic potential of diverse mirna expressions in prostate cancer: A systematic review and meta-analysis
* The Current Landscape of Treatment in Non-Metastatic Castration-Resistant Prostate Cancer
* The multiple roles and therapeutic potential of molecular chaperones in prostate cancer
* Role of chemotherapy in prostate cancer
* StarD13 differentially regulates migration and invasion in prostate cancer cells
- Finally, proofread the paper.

proofread the paper.

Author Response

Point-to-Point Response to Reviewer 4

Thank you very much for providing us with constructive comments on our manuscript. We sincerely appreciate your efforts in helping us to improve our work. We have thoroughly revised the manuscript according to your valuable feedback and have addressed all of your concerns. In addition, we have included new analyses to further strengthen our research. We hope that these adjustments have satisfactorily addressed any issues raised during the review process. All modified parts in the revised manuscript are displayed in red font.

Revision 1: We have edited the manuscript for language by native speaker and made structural adjustments.

Revision 2: We have added background knowledge and identified existing issues in the introduction section, further highlighting the novelty of our research.

Revision 3: We have validated the risk model constructed in this study using an external data set, ICGC.

Revision 4: We have used an optimized machine learning model to add a validation set to assess the impact of risk factors on prognosis.

Revision 5: We have included CCLE data in the expression analysis of the validation factors.

Revision 6: In the discussion section, we have incorporated the latest research advances and provided a future outlook for this field.

Question 1: Paper is well written. Authors should add a little background of the study and limitations of the existing works.

Response:

Thank you very much for your valuable comments and suggestions. We greatly appreciate your feedback, which has significantly improved the quality of our manuscript. We have taken into consideration your recommendations and made revisions to the introduction section by adding background information and discussing the current issues in the field of study (lines 36 to 89 in revised manuscript).

Question 2: Please improve the overall readability of the paper.

Response:

Thank you for your valuable feedback. We highly appreciate your suggestions, which have greatly assisted us in improving the overall structure of our manuscript. We have closely reviewed your comments and made significant revisions to various sections, including the abstract, introduction, and discussion. In addition, we have had our paper professionally edited by a native English speaker to enhance its readability.

Question 3: Numerous solutions to this problem have been proposed in past research. Please explain the novelty and contribution of the study. Why is this study important and what academic values it adds to the field?

Response:

Thank you for your comments. We sincerely apologize for not emphasizing the novelty of our research in the unrevised introduction. We have now revised the introduction and added recent research literature. We have also summarized the innovation of our research as follows, which has been analyzed and modified in detail in the discussion section:

(1) Our study identified a novel biomarker that reflects the malignancy of prostate cancer and constructed a corresponding risk model to predict the survival of prostate cancer patients. Our starting point was the poor prognosis of prostate cancer patients with a GS 9-10, which directly reflects malignant biological behavior. Furthermore, our model reduces the number of prognostic molecules and provides convenient conditions for future clinical application.

(2) Our study utilized machine learning models to determine the impact of risk molecules on prognosis and validated the differential expression of these molecules using single-cell sequencing data sets, CCLE and in vitro experiments. Our findings provide clues for the early clinical diagnosis, identification, and treatment decision-making for prostate cancer, especially high GS prostate cancer. Clinical studies have shown that patients with high GS scores require intensified treatment. If we can use the model to identify and intervene this group, it will improve the overall survival of these patients. Further verification is required by expanding the sample size.

We have made the corresponding revisions, and we appreciate your advice that the innovation of our manuscript was not highlighted in the initial version.

Question 4: Some Paragraphs in the paper can be merged and some long paragraphs can be split into two.

Response:

Thank you for your valuable comments. Your suggestions have been critical in assisting us to adjust the structure of our manuscript. We have revised the manuscript, accordingly, including changes to the abstract, introduction and discussion sections, and have also undergone professional English editing with the aim of improving its readability. For example, in the Discussion section, we have split the original paragraph into smaller sections (lines 421 to 459 in revised manuscript).

We hope that the revised manuscript is now much easier to understand and effectively conveys our research findings. We appreciate your feedback and hope that you find the revised version satisfactory.

Question 5: The paper is a bit too long; the author should try to present the key points to show their contributions while some basic definitions can be shortened to some extent.

Response:

Thank you for your valuable comments. Your suggestions have been critical in assisting us to adjust the structure of our manuscript. We have revised the manuscript, accordingly, including changes to the abstract, introduction and discussion sections, and have also undergone professional English editing with the aim of improving its readability and emphasizing key points.

We hope that the revised manuscript now clearly communicates our research findings and contributes positively to the medical field. We greatly appreciate your feedback and thank you for your continued support throughout this process.

Question 6: What are the limitations of this study?

Response:

Thank you for your questions and comments, which have been extremely valuable in shaping our future work. We appreciate your guidance and look forward to incorporating your suggestions into our research moving forward.

There were various limitations to this study. Firstly, we utilized the TGCA-PRAD and ICGC database for data mining and analysis. Future research requires additional independent data validation and evaluation of the precision of our prognostic model (such as in Peking University First Hospital and other hospitals). Secondly, certain functional molecules within our risk signature contribute significantly to carcinogenesis and malignant tumor development. In vivo and in vitro experiments are necessary to verify the role of these molecules.

Question 7: Authors should summarize the recent works in the form of a table.

Response:

Thank you for your feedback. Your comments have been incredibly helpful in improving the manuscript. Given the extensive research on prostate cancer risk stratification, we have compiled a summary in the Supplementary Materials as Table S3 Research Progress on Risk Stratification and Treatment Outcome Assessment Models for Prostate Cancer. In addition, we have identified some existing issues to help readers quickly understand the current state of research.

Question 8: The contributions of the paper may be highlighted as bullets in the Introduction.

Response:

Thank you for the reviewer's question. We have emphasized the novelty and contribution of our work to current prostate cancer diagnosis and treatment in the Introduction section as follows: In our investigation, we operated the differentially expressed gene (DEGs) analysis of PCa patients and integrated functional enrichment analysis, engineering a prognostic model and OS nomogram based on risk gene signatures to aid clinical decision making, and assessed the correlation between immune infiltration and risk stratification. We aimed to identify the most impactful features among the risk factors for PCa patients’ prognosis by utilizing machine learning techniques in a rigorously written medical journal style for accuracy of expression. Moreover, we validated the model via the International Cancer Genome Consortium (ICGC) dataset. We further used real-time quantitative polymerase chain reaction (RT-qPCR), the single-cell RNA-sequencing (scRNA-seq) datasets, and the human protein atlas (HPA) database to assess the cellular distribution and expression levels for these risk gene signatures. Our study revealed new patterns and perspectives for the risk stratification of PCa patients in current clinical management. Of note, our study has provided a novel perspective for the early clinical recognition of high GS patients and offers novel diagnostic tools for timely intervention in these patients (lines 76 to 89 in revised manuscript).

Response:

Thank you for your question. We have emphasized the contributions of our work in the Introduction section, specifically in providing new insights for the diagnosis, treatment, and clinical interventions for patients with GS 9-10. We believe this is an important contribution to the field and hope that our work will help improve patient outcomes .

Question 9: Authors should discuss and add the below reference in the introduction:

* Clinical theragnostic potential of diverse mirna expressions in prostate cancer: A systematic review and meta-analysis

* The Current Landscape of Treatment in Non-Metastatic Castration-Resistant Prostate Cancer

* The multiple roles and therapeutic potential of molecular chaperones in prostate cancer

* Role of chemotherapy in prostate cancer

* StarD13 differentially regulates migration and invasion in prostate cancer cells

- Finally, proofread the paper.

Response:

Thank you for your valuable feedback. Your suggestions have been extremely helpful in improving the quality of our manuscript. We sincerely apologize for our oversight in missing these important references. We have now added these references to the introduction section (references 7, 8, 9, 10, and 11) to better support our arguments and discussed.

Thank you for your valuable feedback. We have taken your suggestions into consideration and worked with language editing services provided by the MDPI publishing house to improve the quality of our manuscript.

We greatly appreciate your efforts in reviewing our work and understand the importance of constructive criticism in enhancing the quality of our research. Your feedback has been instrumental in guiding us towards the best possible outcome.

Thank you once again for your time and expertise. It has been a pleasure working with you.

Reviewer 5 Report

This paper “Crafting a Personalized Prognostic Model for Malignant Prostate Cancer Patients Using Risk Gene Signatures Discovered through TCGA-PRAD Mining, Machine Learning, and Single Cell RNA-Sequencing”, aims to detect prostate cancer from gene expression analysis on patient samples using The Cancer Genome Atlas (TCGA) multiple gene datasets. The authors exploited an integrated enrichment analysis to inquire the specific functions of these genes. They used Least Absolute Selection and Shrinkage Operator regression method to establish a risk model and evaluate its accuracy through receiver operating characteristic (ROC) curves. Additionally, the authors incorporated clinical variables into a nomogram to predict overall survival. To further examine the impact of risk factor characteristics on Prostate cancer prognosis. The authors dissected the relationship between their prognostic model and immune cell infiltration, and validated their findings using GEO datasets, Prostate cancer cell lines, and tumour tissues from the Human Protein Atlas (HPA) database. Their analysis identified 83 differentially expressed genes in patients among the three cohorts. After enrichment analysis, hormone metabolism and the cell cycle emerged as major drivers of malignant transformation in prostate cancer. Further, WASIR1, KRTAP5-1, TLX1, KIF4A, and IQGAP3 were determined as significant risk factors for overall survival and progression-free survival. Based on these findings, the authors developed a model and nomogram to predict overall survival and progression-free survival, with a C-index of 0.823 (95% CI, 0.766-0.881) and a 10-year area under curve (AUC) value of 0.788 (95% CI, 0.633-0.943). The machine learning model revealed that IQGAP3 and KIF4A were the most influential factors to prognosis. The analysis showed that the established model was interrelated with immune cell infiltration and that the signals were differentially expressed in prostate cancer cells and tissues from single-cell RNA-sequencing datasets.

The topic is justified. The paper could be further improved if the following remarks are taken into consideration:

1.       ABSTRACT: needs to be concise.

2.       Some grammatical mistakes were found in the whole draft of the article; the authors need to fix these.

3.       Introduction section may hold key contribution of this study key folded into it.

4.       Last section of the introduction may hold information related the organization of the rest of the article draft.

5.       The dataset is divided into 8:2 ratio for training and testing but for not validation.

6.       It is unclear from the draft, why nonlinear regression, random forest, and XGBoost is used despite several others state-of-art machine learning models are there in the scikit-learn library, the library of python, authors used for research and experiments.

7.       What was the impact on the results, with 10-fold cross-validation, did authors verified it?

8.       Discussion section needs improvement by comparing with more recent state-of-the-art related studies.

9.       The motivation is not clear. Please specify the importance of the proposed solution.

10.   Discuss the limitations of the proposed method with their possible solutions in the future work section.

11.   Conclusion section may be ‘Conclusion and Future Work’ and must also contain future work-related information.

needs minor grammatical mistakes updations.

Author Response

Point-to-Point Response to Reviewer 5

Thank you very much for providing us with constructive comments on our manuscript. We sincerely appreciate your efforts in helping us to improve our work. We have thoroughly revised the manuscript according to your valuable feedback and have addressed all of your concerns. In addition, we have included new analyses to further strengthen our research. We hope that these adjustments have satisfactorily addressed any issues raised during the review process. All modified parts in the revised manuscript are displayed in red font. Due to the limitations of the system's text box, all modifications made to the images can be referenced in the attached file.

Revision 1: We have edited the manuscript for language by native speaker and made structural adjustments.

Revision 2: We have added background knowledge and identified existing issues in the introduction section, further highlighting the novelty of our research.

Revision 3: We have validated the risk model constructed in this study using an external data set, ICGC.

Revision 4: We have used an optimized machine learning model to add a validation set to assess the impact of risk factors on prognosis.

Revision 5: We have included CCLE data in the expression analysis of the validation factors.

Revision 6: In the discussion section, we have incorporated the latest research advances and provided a future outlook for this field.

The topic is justified. The paper could be further improved if the following remarks are taken into consideration:

Question 1: ABSTRACT: needs to be concise.

Response:

Thank you for your valuable comments. We sincerely apologize for the verbosity of our abstract. Your insights have been immensely helpful in our revision process, and we have since rewritten the abstract and had it professionally edited to meet the highest standards of language proficiency. The revised abstract is as follows:

Background: Prostate cancer is a significant clinical issue, particularly for high Gleason Score (GS) malignancy patients. Our study aimed to engineer and validate a risk model based on the profiles of high GS PCa patients for early identification and the prediction of prognosis. Methods: We conducted differential gene expression analysis on patient samples from The Cancer Genome Atlas (TCGA) and enriched our understanding of gene functions. Using the Least Absolute Selection and Shrinkage Operator (LASSO) regression, we established a risk model and validated it using an independent dataset from the International Cancer Genome Consortium (ICGC). Clinical variables were incorporated into a nomogram to predict overall survival (OS), and machine learning was used to explore the risk factor characteristics' impact on PCa prognosis. Our prognostic model was confirmed using various databases, including single-cell RNA-sequencing datasets (scRNA-seq), the Cancer Cell Line Encyclopedia (CCLE), PCa cell lines, and tumor tissues. Results: We identified 83 differentially expressed genes (DEGs). Further, WASIR1, KRTAP5-1, TLX1, KIF4A, and IQGAP3 were determined to be significant risk factors for OS and progression-free survival (PFS). Based on these five risk factors, we developed a risk model and nomogram for predicting OS and PFS, with a C-index of 0.823 (95% CI, 0.766-0.881) and a 10-year area under the curve (AUC) value of 0.788 (95% CI, 0.633-0.943). Additionally, the 3-year AUC was 0.759 when validating using ICGC. KRTAP5-1 and WASIR1 were found to be the most influential factors of prognosis when using the optimized machine learning model. Finally, the established model was interrelated with immune cell infiltration, and the signals were found to be differentially expressed in PCa cells when using scRNA-seq datasets and tissues. Conclusions: We engineered an original and novel prognostic model based on five gene signatures through TCGA and machine learning, providing new insights into the risk of scarification and survival prediction for PCa patients in clinical practice (lines 11 to 32 in revised manuscript).

Question 2: Some grammatical mistakes were found in the whole draft of the article; the authors need to fix these.

Response:

Thank you for your valuable comments. We sincerely apologize for our oversight regarding some spelling issues. Your insights have been immensely helpful in our revision process, and we have since had the manuscript professionally edited to address these minor grammatical problems.

Question 3: Introduction section may hold key contribution of this study key folded into it.

Response:

Thank you for your valuable comments. Your suggestions have been very helpful in enhancing the innovativeness of our research. We have thoroughly revised the introduction section and made refinements to highlight the innovative aspects of our study (lines 76 to 89 in revised manuscript).

Question 4: Last section of the introduction may hold information related the organization of the rest of the article draft.

Response:

We would like to express our gratitude for your insightful comments, which have proved invaluable in enhancing the innovativeness of our research. Based on your suggestions, we have substantially revised the introduction section and provided refinements at the end of the paragraph to outline the overall concept and methodology of our research, making it easier for readers to comprehend (lines 76 to 89 in revised manuscript).

Question 5: The dataset is divided into 8:2 ratio for training and testing but for not validation.

Response:

Thank you for your valuable feedback. We apologize for our mistakes. We have reanalyzed the data as suggested.

We partitioned TCGA-PRAD patients into training, testing and validation sets in a 6:2:2 ratio. We then utilized machine learning models, including nonlinear regression, random forest, and XGBoost to examine the impact of risk factors on the progression-free interval (PFI) of PCa patients. To optimize model hyperparameters such as maximal number of samples in the leaf and tree depth, we employed RandomizedSearchCV and GridSearchCV from the Python scikit-learn library with 5-fold cross-validation and Bayesian optimization. Additionally, we examined feature importance and interactions using SHAP (Python, version 3.9.2) (lines 147 to 155 in revised manuscript). The figure was shown in Figure 7. We have found that KRTAP5-1 and WASIR1 are the most important factors for prognosis in the output model, which differs from our previous results. We believe this may be due to the increased validation set and the use of Bayesian optimization.

Revised Figure 7:

Figure 7. The importance and interaction of molecular expression feature on prognosis for PCa patients using a machine learning model. (A-B) Comparison of the importance of risk molecular expression level features with the SHAP estimation of randomly generated numbers. (C) Interactive plot. (D) Decision plot.

In addition, we further validated our model using data from the ICGC database and identified 25 patients who met the validation criteria. Therefore, we conducted external validation of the performance of our five-gene risk model. The results showed that in the ICGC cohort, the AUC values of the five-gene risk model for predicting PFS at 2 years, 3 years, and 4 years were 0.942, 0.759, and 0.576, respectively. Due to the limited sample size, we plan to increase the sample size for further validation analysis in the future.

Revised Figure 6:

Figure 6. Prognostic model construction based on risk gene signatures for PCa patients. (A) LASSO regression plots displaying the top five genes. (B) Scatter plot of risk scores ranked from low to high, along with corresponding survival times and status distribution among different PCa samples. Heatmap depicting gene expression in the prognostic model. (C) Kaplan–Meier curves comparing high-risk patients to those with low-risk PCa patients in TCGA-PRAD. (D) Receiver operating characteristic (ROC) curves assessing the ability of this prognostic model to predict progression-free survival (PFS) at one, three, five, and ten years for PCa patients in TCGA-PRAD. (E) Risk scores and survival status chart of patients in the ICGC cohort. (F) Heatmap showing the expression of 5 risk factors in the ICGC cohort. (G) Kaplan–Meier curves comparing high-risk patients to those with low-risk prostate cancer in the ICGC cohort. (H) ROC curves evaluate the prognostic model's ability to predict PFS at 2, 4, and 4 years for PCa patients in ICGC.

Question 6: It is unclear from the draft, why nonlinear regression, random forest, and XGBoost is used despite several others state-of-art machine learning models are there in the scikit-learn library, the library of python, authors used for research and experiments.

Response:

Thank you very much for your comments, they have provided us with great insights. We chose non-linear regression, random forest, and XGBoost models from the scikit-learn library as our basic algorithms, considering their outstanding performance and efficiency in experiments, which provided important help for our research analysis.

At the same time, we have considered using more advanced methods and algorithms in judging the effects of risk factors on prognosis and analyzing their interactions, and we have used Bayesian optimization to reduce model overfitting.

Question 7: What was the impact on the results, with 10-fold cross-validation, did authors verified it?

Response:

Thank you for your valuable comments, which have been very helpful to us. After using LASSO regression analysis to examine the impact of risk factors on the prognosis of prostate cancer, we further used an optimized machine learning model to determine the degree of impact and interaction of these molecules on prognosis, as a parallel analysis.

During the model building process, we used 5-fold cross-validation and Bayesian optimization to prevent overfitting, and to improve the accuracy and robustness of the model. The results showed that WAISR1 and KRTAP5-1 are important factors that affect the prognosis of prostate cancer patients. In addition, the LASSO regression analysis also showed that WASIR1 is the most important factor that affects PFS in prostate cancer patients, suggesting that WAISR1 may play an important role in the survival of prostate cancer patients.

Regarding the validation of differential expression of risk factors, we have only been able to conduct limited testing due to various time constraints. However, we plan to carry out additional testing in the future, including phenotypic and functional validation. We recognize the importance of undertaking testing in all relevant areas, including gene expression, phenotype and functionality assessments.

Question 8: Discussion section needs improvement by comparing with more recent state-of-the-art related studies.

Response:

Thank you for your valuable comments. Your feedback was highly constructive and contributed significantly to the improvement of our discussion section. As per your suggestion, we have revised the discussion section and added a highly relevant research question that is currently popular in building prognosis models: constructing a prognosis model based on key regulatory molecules of programmed cell death (such as disulfidptosis), as well as metabolic processes (lines 427 to 433 in revised manuscript). Our research is based on the direct study of GS 9-10 prostate cancer patients with malignant biological behavior, which is also the innovative aspect of our study.

Question 9: The motivation is not clear. Please specify the importance of the proposed solution.

Response:

Thank you very much for your comments. We apologize for the lack of emphasis on the motivation in our previous manuscript. In response to your feedback, we have revised both the introduction (lines 76 to 89 in revised manuscript) and discussion sections to highlight the purpose and innovation of this study (lines 518 to 525 in revised manuscript).

We appreciate your valuable feedback and have taken it into consideration for improving the quality of our manuscript.

Question 10: Discuss the limitations of the proposed method with their possible solutions in the future work section.

Response:

Thank you for your valuable suggestions. Your comments are constructive, and we appreciate your input. We have taken your feedback into consideration and in the discussion and conclusion section, we have outlined the limitations of our current research and proposed solutions for future studies (lines 518 to 525 in revised manuscript). And future research requires additional independent data validation and evaluation of the precision of our prognostic model (such as in Peking University First Hospital and other hospitals). We also are preparing to conduct functional experiments.

Question 11: Conclusion section may be ‘Conclusion and Future Work’ and must also contain future work-related information.

Response:

Thank you for your feedback. Your input has been extremely helpful, and we appreciate your time reviewing our manuscript. As per your suggestion, we have updated the section title to "Conclusion and Future Work".

Thank you for your valuable feedback. We have taken your suggestions into consideration and worked with language editing services provided by the MDPI publishing house to improve the quality of our manuscript.

We greatly appreciate your efforts in reviewing our work and understand the importance of constructive criticism in enhancing the quality of our research. Your feedback has been instrumental in guiding us towards the best possible outcome.

Thank you once again for your time and expertise. It has been a pleasure working with you.

Round 2

Reviewer 1 Report

I commend the authors for adding a substantial amount of work. The manuscript has improved. My remaining concerns:

Figure 6. ICGC cohort. I am glad to see positive results. For the TCGA cohort patients with risk score < 1 are classified as low risk. For the ICGC cohort patients with risk score s < ~6 are classified as low risk. How do the risk scores relate to each other across these two cohorts?

Heatmap Fig 6F should be z-scaled row-wise as the emphasize relative expression changes. Just like the authors did in Fig 6B. Colors scheme (dead/alive) should be consistent between Fig 6B and E.

Fig 6D. AUC for 10-years is greater in text (0.788) compared to 1-year (0.765). However, in plot blue seems to have the greater AUC. Can the authors explain this?

Fig 9. The authors have followed my suggestion to create a boxplot of the expression of the signature genes. However, visualization parameters are set such that it is impossible to recognize any expression patterns. Indeed, it looks like genes are expressed in < 1% of cells. I cannot recognize the authors following statement: “It is worth noting that while the expression levels of the five risk factors mentioned in this dataset were relatively low, they still exhibited distinct distribution patterns. Specifically, KIF4A was found to be expressed in tumor cells, T cells, monocytes, and fibroblasts. IQGAP3 was expressed in tumor cells, T cells, and monocytes, while WASIR1 was primarily expressed in mast cells”

Author Response

Point-by-Point Response to Reviewer 1

Thank you very much for providing us with constructive comments on our manuscript. We sincerely appreciate your continuous efforts in helping us to improve our work. We have thoroughly revised the manuscript according to your valuable feedback and have addressed all your concerns. In addition, we have included new analyses to further strengthen our research. We hope that these adjustments have satisfactorily addressed any issues raised during the review process. All modified parts in the revised manuscript are displayed in red font. Due to the limitations of the system's text box, all modifications made to the images can be referenced in the attached file.

Revision 1: We have revised the color schemes of the testing and validation cohort in Figure 6 to make it more consistent.

Revision 2: We have revised the ROC curve in Figure 6D by removing the smoothing process.

Revision 3: we have added a supplementary Figure 1S to showcase the expression profiles of risk molecules across different cell subpopulations. This figure is presented alongside the dot plot in Figure 9.

I commend the authors for adding a substantial amount of work. The manuscript has improved. My remaining concerns:

Question 1: Figure 6. ICGC cohort. I am glad to see positive results. For the TCGA cohort patients with risk score < 1 are classified as low risk. For the ICGC cohort patients with risk scores < ~6 are classified as low risk. How do the risk scores relate to each other across these two cohorts?

Response:

Thank you for your valuable comments, which are extremely helpful for our research. Your question about the selection of cutoff values for different queues in the testing and validation sets is an excellent one and is a common issue in this type of research.

In our study, the cutoff values in the TCGA database are around 1, while the cutoff values in the ICGC database are around 6. The ICGC's risk score is calculated from a constructed risk formula from TCGA, and the optimal cutoff is calculated and used to ensure the difference of the risk model in ICGC. Due to the limited number of eligible patients in the ICGC database (N = 25), the calculated cutoff value differs from that of TCGA. While it is principle to use the same cutoff value for the same dataset, the platforms and baselines of sequencing, as well as the heterogeneity of data may vary across databases. Therefore, different cutoff values have been calculated in testing and validation sets in some studies (see references for actual examples).

We greatly appreciate your helpful questions regarding our study quality control, which is limited by the small number of eligible patients in the ICGC database. We plan to expand our dataset and adopt the same platform, quality control, and data processing methods as the TCGA database to ensure the same cutoff value and validate the model's efficacy.

References:

Xu J, Zhang K, Zhang G. Prognostic Lysosome-Related Biomarkers for Predicting Drug Candidates in Hepatocellular Carcinoma: An Insilco Analysis. J Hepatocell Carcinoma. 2023;10:459-472. Published 2023 Mar 21. doi:10.2147/JHC.S401338

Hu, D.; Messadi, D.V. Immune-Related Long Non-Coding RNA Signatures for Tongue Squamous Cell Carcinoma. Curr. Oncol. 2023, 30, 4817–4832. https:// doi.org/10.3390/curroncol30050363

Question 2: Heatmap Fig 6F should be z-scaled row-wise as the emphasize relative expression changes. Just like the authors did in Fig 6B. Colors scheme (dead/alive) should be consistent between Fig 6B and E.

Response:

Thanks for your comments. I would like to express my gratitude for the reviewer's suggestions which were helpful in improving the quality of our manuscript. Your persistent pursuit of scientific research is worthy of our admiration and serves as a valuable learning opportunity. We acknowledge that we have not achieved a consistent color scheme for the figures in our manuscript. Following your recommendations, we have now unified the image style of TCGA and ICGC cohort analysis, resulting in a more organized display.

Revised Figure6:

Figure 6. Prognostic model construction based on risk gene signatures for PCa patients. (A) LASSO regression plots displaying the top 5 genes. (B) Scatter plot of risk scores ranked from low to high, along with corresponding survival times and status distribution among different PCa samples. Heatmap depicting gene expression in the prognostic model. (C) Kaplan–Meier curves comparing high-risk patients to those with low-risk PCa patients in TCGA-PRAD. (D) Receiver operating characteristic (ROC) curves assessing the ability of this prognostic model to predict progression-free survival (PFS) at 1, 3, 5, and 10 years for PCa patients in TCGA-PRAD. (E) Scatter plot of risk scores ranked from low to high, along with corresponding survival times and status distribution among different PCa samples. Heatmap depicting gene expression in the prognostic model in ICGC. (F) Kaplan–Meier curves comparing high-risk patients to those with low-risk prostate cancer in the ICGC cohort. (G) ROC curves evaluate the prognostic model's ability to predict PFS at 2, 4, and 4 years for PCa patients in ICGC.

Question 3: Fig 6D. AUC for 10-years is greater in text (0.788) compared to 1-year (0.765). However, in plot blue seems to have the greater AUC. Can the authors explain this?

Response:

Thank you for your feedback. Your comments have been very helpful in improving the overall quality of our manuscript. We previously used a smoothing processing for the ROC curve in Figure 6D, which made the 10-year AUC look less impressive compared to the 1-year AUC. Based on your suggestion, we have removed the smoothing and used the same ROC curve as Figure 6G, which will avoid any misunderstandings. Thank you again for your valuable input.

Revised Figure 6D:

(D) Receiver operating characteristic (ROC) curves assessing the ability of this prognostic model to predict progression-free survival (PFS) at 1, 3, 5, and 10 years for PCa patients in TCGA-PRAD.

Question 4: Fig 9. The authors have followed my suggestion to create a boxplot of the expression of the signature genes. However, visualization parameters are set such that it is impossible to recognize any expression patterns. Indeed, it looks like genes are expressed in < 1% of cells. I cannot recognize the authors following statement: “It is worth noting that while the expression levels of the five risk factors mentioned in this dataset were relatively low, they still exhibited distinct distribution patterns. Specifically, KIF4A was found to be expressed in tumor cells, T cells, monocytes, and fibroblasts. IQGAP3 was expressed in tumor cells, T cells, and monocytes, while WASIR1 was primarily expressed in mast cells.”

Response:

Thank you for your advice. Your suggestions have been very helpful in improving the quality of our manuscript. Indeed, we have observed relatively low expression of risk genes in the presented dataset. Therefore, we have created a Supplementary Figure 1 showing the expression of risk signatures in different cell subpopulations. This figure is now uploaded along with the original dotplot from Figure 9C, which we have combined for a joint analysis (as shown in Figure 9).

Figure S1:

Figure S1. (A) The annotation of cell subpopulations. Analysis of the expression of KIF4A (B), IQGAP3 (C), and WASIR1 (D) in cellular subgroups using the GSE 141445 dataset.

We have addressed this issue in our manuscript by providing a more moderate description of the low expression of risk-associated molecules within subpopulations of single cells, and outlining measures and strategies for future improvement. Specifically, we have described that in our manuscript we have added descriptions indicating low expression of these five molecules within subpopulations of single cells, and outlined measures and strategies to improve upon this issue in the following manner:

To clarify the expression profiles of various cell types within the PCa tissue, we performed an analysis on the GSE 141445 dataset. As depicted in Figure 9A, our cell grouping results were presented along with markers identified by the clustering method shown in Figure 9B. It is worth noting that while the expression levels of the five risk factors mentioned in this dataset were relatively low, they still exhibited different distribution patterns. Specifically, KIF4A was found to be expressed in tumor cells, T cells, monocytes, and fibroblasts. IQGAP3 was expressed in tumor cells, T cells, and monocytes, while WASIR1 was primarily expressed in mast cells (Figure 9C and Supplementary Figure 1). Moving forward, as single-cell RNA sequencing methods continue to improve, we plan to integrate additional datasets to further refine our understanding of the expression distributions of these important risk factors (lines 362 to 372 in revised manuscript).

We also added corresponding discussion and analysis to the discussion section, as follows:

Secondly, the risk factors are expressed at a low level in the scRNA-seq dataset, and with the maturity of scRNA-seq technology, more datasets can be analyzed in the future (lines 516 to 518 in revised manuscript).

We have analyzed the reasons for the low expression of risk signatures in different cell subpopulations. The reasons may include the PCa tissue type and location, the number and efficiency of captured cells, the patient's pathological grade, the markers used for cell labeling, the sparse nature of sing-cell RNA-sequencing (scRNA-seq) datasets as well as a high noise level. In addition, non-expressed genes and technical shortcomings, such as dropout events (unsequenced transcripts), result in many zeros in the expression matrix, and an incomplete description of a single cell's transcriptome. These are current scientific and technical issues that need to be addressed in interpreting phenotypic and molecular expression using single-cell sequencing.

Besides, we believe that heterogeneity is a universal issue in current stages of single-cell sequencing technology, and it has been observed in many single-cell sequencing studies. Therefore, we adopted multiple methods to verify and validate the expression of risk-associated molecules in the interpretation and conclusion of our research results, ensuring the reliability and general applicability of our findings.

In this study, we included information from 36,424 cells, while the TCGA database provided RNA-seq data on bulk tumor samples, meaning that scRNA-seq data may not fully represent the expression profiles of solid tumors. As the scRNA-seq technology continues to mature and improve, with the application of models and methods such as MAGIC, PAGODA, and SINCERA, the number of cells analyzed will increase, and more scRNA-seq datasets will become available. We also plan to continue analyzing the expression of these risk signatures in single cells in the future.

References:

Chen S, Zhu G, Yang Y, et al. Single-cell analysis reveals transcriptomic remodellings in distinct cell types that contribute to human prostate cancer progression. Nat Cell Biol. 2021;23(1):87-98. doi:10.1038/s41556-020-00613-6

Lafzi A, Moutinho C, Picelli S, Heyn H. Tutorial: guidelines for the experimental design of single-cell RNA sequencing studies. Nat Protoc. 2018;13(12):2742-2757. doi:10.1038/s41596-018-0073-y

We greatly appreciate your efforts in reviewing our work and understand the importance of constructive criticism in enhancing the quality of our research. Your feedback has been instrumental in guiding us towards the best possible outcome.

Thank you once again for your time and expertise. It has been a pleasure working with you.

Round 3

Reviewer 1 Report

The authors have done a great job addressing all of my concerns.